# Antibody conjugates for targeted delivery of Toll-like receptor 9 agonist to the tumor tissue

**Diana Corogeanu**[1¤], **Kam Zaki**[2], **Andrew J. Beavil**[3], **James N. Arnold**[3], **Sandra S. Diebold**[1]*

**1** National Institute for Biological Standards and Control (NIBSC), Biotherapeutics Division, Medicines and Healthcare products Regulatory Agency, Potters Bar, United Kingdom, **2** National Institute for Biological Standards and Control (NIBSC), Advanced Therapies Division, Medicines and Healthcare products Regulatory Agency, Potters Bar, United Kingdom, **3** King's College London, School of Cancer and Pharmaceutical Sciences, Faculty of Life Sciences and Medicine, Guy's Hospital, London, United Kingdom

¤ Current address: Conquest Hospital, Hastings, United Kingdom
* sandra.diebold@nibsc.org

**Data Availability Statement:** The data will be included as a Supporting information file in form of an excel spreadsheet.

**Funding:** James Arnold is supported by a grant from Cancer Research UK (DCRPGF\100009) and is the recipient of a Cancer Research Institute /

## Abstract

Imiquimod, a Toll-like receptor 7 (TLR7) agonist is routinely used for topical administration in basal cell carcinoma and stage zero melanoma. Similarly, the TLR agonist Bacillus Calmette-Guérin is used for the local treatment of bladder cancer and clinical trials showed treatment efficacy of intratumoral injections with TLR9 agonists. However, when administered systemically, endosomal TLR agonists cause adverse responses due to broad immune activation. Hence, strategies for targeted delivery of TLR agonists to the tumor tissue are needed to enable the widespread use of endosomal TLR agonists in the context of tumor immunotherapy. One strategy for targeted delivery of TLR agonist is their conjugation to tumor antigen-specific therapeutic antibodies. Such antibody-TLR agonist conjugates act synergistically by inducing local TLR-mediated innate immune activation which complements the anti-tumor immune mechanisms induced by the therapeutic antibody. In this study, we explored different conjugation strategies for TLR9 agonists to immunoglobulin G (IgG). We evaluated biochemical conjugation of immunostimulatory CpG oligodesoxyribonucleotides (ODN) to the HER2-specific therapeutic antibody Trastuzumab with different cross-linkers comparing stochastic with site-specific conjugation. The physiochemical make-up and biological activities of the generated Trastuzumab-ODN conjugates were characterized *in vitro* and demonstrated that site-specific conjugation of CpG ODN is crucial for maintaining the antigen-binding capabilities of Trastuzumab. Furthermore, site-specific conjugate was effective in promoting anti-tumor immune responses *in vivo* in a pseudo-metastasis mouse model with engineered human HER2-transgenic tumor cells. In this *in vivo* model, co-delivery of Trastuzumab and CpG ODN in form of site-specific conjugates was superior to co-injection of unconjugated Trastuzumab, CpG ODN or stochastic conjugate in promoting T cell activation and expansion. Thereby, this study highlights that site-specific conjugation of CpG ODN to therapeutic antibodies targeting tumor markers is a

Wade F.B. Thompson CLIP grant (CRI3645). The funders had no role in study design, data collection an analysis, decision to publish, or preparation of the manuscript.

**Competing interests:** The authors have declared that no competing interests exist.

feasible and more reliable approach for generation of conjugates which retain and combine the functional properties of the adjuvant and the antibody.

## Introduction

TLR3, TLR7/8 and TLR9 are pattern recognition receptors that reside intracellularly, in the endosome of innate immune cells including antigen-presenting cells such as dendritic cells (DC) and macrophages. These endosomal TLR detect nucleic acids from viruses and other intracellular pathogens and mediate innate immune activation. Upon activation by endosomal TLR, DC undergo maturation, migrate to the draining lymph nodes and present exogenous antigens to CD4 and CD8 T cells. DC activated by the detection of nucleic acid agonists via endosomal TLR instruct adaptive immune responses towards a T helper 1 (Th1) phenotype with strong cytotoxic T cell (CTL) effector functions for elimination of infected cells [1]. CTL responses, vital for anti-viral immunity, are also thought to be a crucial mechanism for the eradication of tumor cells in response to immunotherapy [2]. Hence, agonists of endosomal TLR have been explored in the context of cancer therapy and were demonstrated to promote anti-tumor immune responses when administered to the tumor site in experimental and clinical settings [3]. The TLR7 agonist imiquimod is routinely used for the topical treatment of basal cell carcinoma and a wide range of TLR agonists are explored in clinical trials for cancer treatment [4]. However, upon systemic administration endosomal TLR agonists induce broad immune activation that leads to adverse reactions, which are obstructive for their wider use apart from topical administration. Strategies for targeted delivery of TLR agonists to the tumor tissue may overcome these limitations.

Antibodies are suitable for targeted delivery of cytotoxic payloads to tumor cells as demonstrated by a number of licenced antibody-drug conjugates (ADC) such as Gemtuzumab ozogamicin, Brentuximab vedotin and Trastuzumab emtansine [5]. Antibodies can be similarly used to deliver endosomal TLR agonists to the tumor site in order to induce a local inflammatory microenvironment that promotes an effective anti-tumor immune response. Antigen-presenting cells such as macrophages and DC residing in or around the tumor tissue have the ability to take up antibody-coated tumor cells and thereby can access the antibody-TLR agonist conjugate and the tumor-associated antigens together. By concentrating endosomal TLR agonists in the tumor site through targeted antibody-mediated delivery, innate immune activation in normal tissue should be kept to a minimum thereby avoiding or limiting widespread systemic adverse responses.

In order to compare stochastic with site-specific conjugation of antibody to TLR agonists, we used the anti-HER2/neu specific therapeutic antibody Trastuzumab for covalent conjugation to CpG ODN which trigger TLR9-mediated immune activation. We selected this combination of antibody and nucleic acid TLR agonist since Trastuzumab is a monoclonal antibody with established clinical efficacy in cancer treatment and CpG ODN are more stable than RNA oligoribonucleotides (ORN) used as TLR7 agonists and are molecular defined unlike polymeric heterogeneous mixtures such as the TLR3 agonist polyI:C. Furthermore, we had prior experience in generating antibody-ODN conjugates for DC vaccination [6].

We compared conjugates generated with the heterobifunctional crosslinker succinimidyl-4-(N-maleimidomethyl) cyclohexane-1-carboxylate (SMCC) or cyanuric chloride (CC) and generated conjugates of Trastuzumab by either stochastic conjugation or by site-specific crosslinking via engineered reactive cysteine residues using ThioMab technology [7, 8]. The

generated Trastuzumab-ODN conjugates were tested *in vitro* to compare their HER2-binding activity, their tumor cytotoxic function and their adjuvanticity to that of unconjugated Trastuzumab and CpG ODN. Furthermore, we engineered B16 melanoma cells expressing human HER2 to evaluate the anti-tumor immune response induced by the conjugates in a pseudo-metastasis mouse model.

## Material and methods

### Ethics statement

All animal experiments were undertaken in accordance with UK governmental regulations (Animal Scientific Procedures Act 1986) under the project licence PPL 70/8831. Ethics approval for the *in vivo* study was granted by the AWERB committee at NIBSC.

### Antibodies and oligonucleotides

We used Trastuzumab that was either generated inhouse or its commercial equivalent Herceptin® (Roche, Basel, Switzerland). Trastuzumab and its commercial form Herceptin® have an identical amino acid sequence [9] but may harbour a different glycosylation profile. However, inhouse generated antibody behaved similarly in all our assays to the commercial form of the antibody. We, therefore, used Trastuzumab and its commercial form Herceptin® interchangeably for the physiochemical and *in vitro* characterization of conjugates. In mouse *in vivo* experiments, we used exclusively Trastuzumab for direct comparison to inhouse produced Thiomab-modified antibody.

For the expression of Trastuzumab inhouse, we used the pVITRO-Trastuzumab IgG1/κ plasmid from Addgene (#61883) as vector [10]. Antibodies were generated in Freestyle™ 293T cells (Life Technologies Limited) by transient transfection. Supernatants were screened for presence of IgG1 antibody by sandwich ELISA (purified capture antibody: clone G17-1; biotinylated detection antibody: clone G18-145; both BD Biosciences) and antibody was purified by FPLC in an AKTA Start 29022094 with Unicorn Start 1.0 software using a Protein A HiTrap MabSelect SuRe column (GE Healthcare). Upon elution of the antibody the buffer was exchanged against PBS using Amicon Ultra centrifugation filters with either 10kDa or 100kDa cut-off. Antibody yield was determined by BCA assay (Pierce). Isotype control antibody and ThioMAb antibodies were expressed and purified using the same protocol.

For generation of Trastuzumab antibody for site-specific conjugation we employed Thio-MAb technology. The nucleotides encoding valine (codon GTC) at position 205 in the constant region of the Trastuzumab light chain were exchanged with nucleotides encoding cysteine (codon TGT) by PIPE cloning using a three fragment cloning approach (pVITRO F1 `CTTTTGAGCGGAGCTAATTCTCGGG` and LV205C.rv01 `ATCAGGGCCTGAGCTCGCCCTGTA CAAA`; LV205C.fw01 `GCCCTGTACAAAGAGCTTCAACAG` and ratIgG2bmodif.rv `ATCCCTAA TACCTGCCACC`; ratIgG2b.fw `TATCCCTAATACCTGCCACCCCACTC` and pVITROrvF1 `CTTGAGTTTTGAGCGGAGCTAATTCT`) [11].

Isotype control antibody and the corresponding ThioMAb were generated by PIPE cloning and the suitability of the antibodies as isotype controls was confirmed by flow cytometry. The heavy chain variable region of the antibody was cloned from an antibody specific for a plant protein, while the light chain was derived from a recombinant antibody with a negative staining profile. Both variable regions were cloned into the Trastuzumab IgG1/κ or the Trastuzumab ThioMab plasmid backbone replacing the Trastuzumab-specific variable regions. The resulting isotype control and the isotype ThioMab antibodies both showed negative staining on human PBMC, mouse splenocytes, B16 cells and B16-derived cell lines expressing human HER2.

Unmodified and modified phosphorothioate immunostimulatory ODN CpG 1668 (TCCATGACGTTCCTGATGCT) and non-stimulatory control ODN GpC 1668 (TCCAT GAGCTTCCTGATGCT) were purchased from MWG Eurofins (Ebersberg, Germany). Modified ODN with a 3' thiol modification (S-CpG 1668 and S-GpC 1668) were used for conjugate generation.

## Generation of antibody-ODN conjugates

For conjugation, SMCC was added to the antibody at 5–30 molar excess and the mixture was incubated at 22ºC for 1h. Samples were purified from excess unreacted SMCC by ultracentrifugation using 0.5ml Amicon Ultra filters and were recovered in a small volume (30–50μl). In parallel, S-CpG or S-GpC was reduced in 50mM DTT for 1 h at 22ºC. CpG was purified from DTT using Micro Bio-Spin™ 6 Chromatography Columns. Reduced and purified S-CpG or S-GpC was added at 3–10 molar excess to SMCC-activated antibody and samples were incubated for 2h at 22ºC. Antibody-conjugates were purified from unreacted CpG by ultrafiltration using Amicon filters with a 100kDa cut-off. Samples were recovered in a small volume of PBS and sterile filtered (pore size 0.22μm).

For conjugation with CC, 300μg CC in acetone was added to a glass vial and dried. 300μg S-CpG or S-GpC in borate buffer at a concentration of 1mg/ml was added to the glass vial and the solution was stirred at RT for 90min. Unreacted CC was removed using Diethyl ether extraction. Next, 150μl of antibody in borate buffer at 2mg/ml was added to the CC-reacted CpG or GpC ODN. The mixture was incubated at 37ºC overnight. The next day, samples were purified from unreacted ODN using Amicon filters with a 100kDa cut-off. As for SMCC conjugates, samples were recovered in a small volume of PBS and sterile filtered.

For ThioMab conjugates, we reduced the antibody in TCEP to remove unwanted adducts bound to the antibody via disulphide bonds to thiol-containing amino acids, then purified the resulting heavy and light chains by ultrafiltration. In a following step, heavy and light Ig chain disulphide bonds were reoxidated using the accelerator haloalkane dehalogenase. The engineered cysteines of TH-Trastuzumab are surrounded by positively charged amino acids and hence reoxidize at a much lower rate, remaining available for conjugation in the used protocol. CpG ODN synthesized with an amine group at the 3' end was treated with the cross-linker SMCC to generated reactive maleimide groups for conjugation to reoxidized TH-Trastuzumab [12].

The protein concentration of conjugates was analyzed by BCA assay (Pierce) whereas the nucleic acid concentration was determined by spectrophotometry. A standard curve was generated containing 0.5μg/μl of human IgG1 antibody combined with increasing concentrations of ODN (6.25–200μg/ml ODN) in PBS. The absorbance at 260nm for the standard curve was measured in the Nanodrop in triplicates. The concentration of conjugate samples was adjusted to 0.5μg/μl protein content before absorbance at 260nm was determined in a Nanodrop 1000 spectrophotometer. The molar ratio of CpG to antibody was calculated accordingly for each conjugate preparation.

## Physicochemical characterization of conjugates

For size exclusion chromatography (SEC-HPLC) analysis, 10μg of either antibody or antibody-ODN conjugate was diluted in 0.2M Sodium Phosphate, 0.1 M Sodium Sulphate pH 6.0 and left at 6˚C for 30min to allow equilibration between monomeric and dimeric material. The TSK G4000SWXL column was equilibrated in 0.2 M Sodium Phosphate, 0.1M Sodium Sulphate pH 6.0 and maintained at 25ºC under constant pressure (14bar). 10μl water were run

to establish the baseline. Then, the prediluted analytes were injected. Flow rate was 0.4ml/min and run time was 60min.

Samples were analyzed by SDS-Page analysis on Novex Wedge 4–20% gradient electrophoresis gels (Life Technologies) under non-reducing or reducing conditions. As markers ColorPlus pre-stained Protein Standard and Mark12 Unstained Standard were used. To visualise the ODN component of the conjugates, the gel was first stained for nucleic acid with 2µg/ml Ethidium Bromide in water for 30min. Subsequently, the gel was destained in water and imaged with the Syngene PXi System. Next, the gel was stained for protein for 1h in SymplyBlue SafeStain. The gel was destained for 1h in water or overnight in 2% NaCl and imaged using the Coomasie Blue channel with the Syngene PXi System. Images were analyzed with GeneTools from Syngene software.

The antibody-ODN conjugate preparations were tested in the gel-clot method of the LAL assay using Endosafe lysate from Charles River Laboratories and the 3rd WHO International Standard (10/178) for endotoxin quantification. The endotoxin concentration of the antibody preparations was below 0.05IU/ml.

## Functional *in vitro* characterization of conjugates

The ability of Trastuzumab and Trastuzumab-ODN conjugates to arrest cell growth of tumor cells was tested *in vitro*. Human HER2 positive or negative cell lines and B14.3 HER2 cell lines were seeded at $1x10^4$ cells per well in 96-well plates. Serial dilutions of vehicle control, isotype control, ODN, Trastuzumab and Trastuzumab-ODN conjugates were added and cells were incubated at 37ºC, 5% $CO_2$ for 48h. Next, 20µl of the CellTiter 96® AQueous One Solution Reagent containing the MTS (3-(4,5-dimethylthiazol-2-yl)-5-(3-carboxymethoxyphenyl)-2-(4-sulfophenyl)-2H-tetrazolium, inner salt) compound was added per well. Plates were left at 37ºC, 5% CO2 for 30min and plates were analyzed in a SpectraMax M5 96 well plate reader (Molecular Devices) measuring absorption at 490nm. OD490 of vehicle-treated cells was considered 100% proliferation in the conditions of the assay. Proliferation of treated cells was expressed as percent of vehicle-treated cells.

For the ADDC assay, BT474 cells were labelled with 11.2µM calcein in RPMI 1640 medium containing 5% heat-inactivated FCS for 30min at 37˚C and 5% $CO_2$. Cells were washed and resuspended at $1.6x10^5$ cells/ml in complete RPMI 1640 medium containing 10% heat-inactivated FCS. $8x10^3$ cells were plated per well in a 96-well plate. Trastuzumab, conjugates or controls were diluted in complete medium and added to the BT474 cells. Samples were prepared in triplicates and the assay plate was incubated for 30min at 37˚C and 5% $CO_2$. Cryo-preserved PBMC, which has been isolated by density gradient centrifugation, were thawed and added to the calcein-loaded target cells at an effector: target cell ratio of 25:1 and the cultures were incubated overnight at 37˚C and 5% $CO_2$. The next day, lysis buffer was added to the wells containing the maximum release control. After a 30min incubation, the cell supernatants were harvested and added to a white bottomed 96-well plate. Fluorescence was read using excitation at 494nm and emission at 517nm in a spectrophotometer. A dose response corrected for spontaneous calcein release from target cells was calculated using the following formula: % cytotoxicity = (experimental value—target cell spontaneous control) / (target cell maximal release—target cell spontaneous control).

To compare the binding properties of antibody-ODN conjugates with unconjugated antibody, human HER2 or human EGFR (both Sino Biological) was immobilised on Nunc Maxisorp Immunoplates at 0.5 and 1µg/ml in PBS, respectively. Upon coating, plates were washed and blocked with PBS containing 2.5% heat-inactivated FCS and 0.02% Sodium Azide. Upon washing of the plates, antibody and conjugate samples were added starting a top concentration

of 500ng/ml and performing 1:2 serial dilutions. The ELISA plates were incubated at 4ºC and developed the next day. Plates were incubated with anti-human IgG1 detection antibody (clone G18-145, BD Biosciences) followed by incubation with ExtrAvidin-Alkaline Phosphatase and subsequently SIGMAFAST™ pNPP substrate (Sigma-Aldrich). Alternatively, plates were washed and incubated with HRP anti-human kappa light chain antibody followed by addition of TMB substrate. For the latter, the reaction was stopped by addition of 2M sulfuric acid. Plates were measured for absorption in a SpectraMax M5 96 well plate reader (Molecular Devices).

The immune stimulatory activity of CpG 1668 ODN in soluble or conjugated form was evaluated *in vitro* using mouse GM-CSF BMDC. $1$-$5$x $10^5$ BMDC were seeded per well in TC-treated 96-well plates. ODN, antibody and antibody-ODN conjugates were added to the wells at different concentrations. For co-cultures with tumor cells, B14.3 HER2 cells were seeded in TC-treated 96-well plates and incubated with ODN, antibody or antibody-ODN conjugates for 30min prior to addition of $1$-$5$x $10^5$ BMDC per well at a BMDC: tumor cell ratio of 2:1. Samples were performed in triplicates. Plates were incubated overnight at 37ºC, 5% $CO_2$. The IL-6 and IL-12p40 cytokine levels in the supernatant were measured by sandwich ELISA (IL-6: purified capture antibody clone MP5-20F3 and biotinylated detection antibody clone SXC-1; IL-12 p40: purified capture antibody clone C15.6 and biotinylated detection antibody clone C17.8).

## Generation of B16-derived cell line expressing human HER2

The sequence for HER2 was subcloned from pcDNA3-HER2WT (Addgene plasmid #16257) into the transgene pSIN-SFFV vector by PIPE cloning using a three-fragment approach (pSIN-HER2_fw CTGAGTCGCCCGGGGGATGGAGCTGGCGGCCTTGTGCC and pSIN-HER2_rv CTGGGTCTGGACGTGCCAGTGTGAGCGGCCGCGACTCTAG; pSINSFFV_fw ACGTCCAGGCACGTATTGTGATGAGCGA and pSIN-fusion_rv ACAGACT–GAGTCGCCCGGGGG; pSIN-fusion_fw GCGGCCGCGACTCTAGAGTCGA and pSINSFFV_rv ACGTCCAGGCACGTATTGT) [13]. For lentivirus production, HEK293T cells were transiently co-transfected with plasmids pSIN-SFFV-HER2, p8.91 and pMD.G at 1.5:1:1 ratio in Opti-MEM (Gibco) using FuGENE (Promega) transfection reagent. Supernatants containing lentivirus were harvested after 48-72h, filtered through a 0.45μm syringe filter and concentrated 200-fold by ultra-centrifugation.

The B16-derived cell line B14.3 expressing an ovalbumin (OVA)-GFP fusion protein was transduced with human HER2-encoding lentivirus at multiplicity of infection (MOI) of 1, 10 and 20 [6, 14]. At day 7 post-transduction, cells were stained for HER2 expression and analyzed by flow cytometry. Samples showing a distinct HER2 positive population were subjected to fluorescence activated cell sorting using a BD FACS Aria II cell sorter with a 100μM nozzle sorting single cells into 96-well plates. Single-cell clones were selected for robust cell growth and expression of human HER2 and OVA-GFP fusion protein. Tumor growth profiles in C57BL6 mice were conducted for a few clones to identify the most suitable cell line named B14.3 HER2.

## Tumor model

C57BL/6 mice were purchased from Envigo (Blackthorn, UK). The animals received a combination of the following procedures: Mice were injected into the tail vein with $0.5$–$0.9$x $10^5$ B16 or B14.3 HER2 cells in 150μl PBS [14]. Mice were treated with Trastuzumab, isotype control antibody or antibody conjugates on days 4, 8 and 11 post tumor inoculation by intravenous injection. Blood samples were harvested at day 1, 14 and 18 post tumor inoculation. 15–18

days after tumor inoculation, all mice were culled by $CO_2$ asphyxiation followed by spinal cord dislocation and blood, lungs and spleens were harvested. Tumor volume was not determined in this melanoma pseudo-metastasis model. Instead, the number of tumor nodules on the removed lungs was determined under a binocular. Mice were anaesthetized using isoflurane and subsequently received mashed food to support hydration. Throughout the procedures mice experienced only mild severity with less than 1% of animals showing moderate severity levels and additional measures to alleviate suffering were not required.

Lungs were weighted and either fixed in Fekete solution for subsequent tumor nodule enumeration or single cell suspensions of PBS-perfused lungs were generated using the gentle-MACS™ tissue dissociator. Splenocyte suspensions were generated upon digestion with DNase I and collagenase. Single cell suspensions of splenocytes and lungs were phenotypically analyzed by flow cytometry after staining with LIVE/DEAD Fixable Violet Dead Cell Stain (Thermo Fisher Scientific) and a combination of the following fluorescently labelled antibodies: anti-CD3, anti-CD4, anti-CD8α, anti-CD44, anti-CD45, anti-CD62L and Trastuzumab antibody. Staining for surface markers was either performed on live cells or cells fixed in 50% (v/v) of Streck solution. All samples were analyzed on 3-laser FACS Canto II or 3-laser LSR Fortessa flow cytometry instruments (BD Bioscience) and data files were analyzed using FlowJo v10 software.

Splenocytes isolated from tumor-bearing mice were stained with 2μM CFSE for 10 min at 37ºC. Cells were washed twice before restimulating 1-2x $10^6$ splenocytes in the presence of 2-4x $10^5$ GM-CSF BMDC and antigen mix in 24-well plates. The antigen mix contained recombinant human HER2, the OVA class I-restricted peptide SIINFEKL and several predicted TRP1- and TRP2-specific peptides. Cells were harvested after 72h and stained for live/dead-cell discrimination using a fixable dye (Invitrogen). Upon fixation, cells were stained for surface marker analysis of CD3, CD4, CD8 and the number of proliferating CFSE-positive cells was analyzed by flow cytometry.

## Statistics

GraphPad Prism was used for statistical analysis. Data are shown as mean ± standard deviation (SD). Data was analyzed using an appropriate Student's T test for each set, as specified for each figure. Differences were considered significant when $p \leq 0.05$.

## Results

### Conjugate generation

First, Trastuzumab was conjugated to CpG ODN using the heterobifunctional cross-linker SMCC. This conjugation strategy was employed for generation of the ADC Trastuzumab emtansine (Kadcyla®), approved for clinical use [15]. Furthermore, our lab used this crosslinker to successfully generate antibody-CpG ODN conjugates for DC targeting [6].

Upon generation of Trastuzumab and Isotype control conjugates using titrations of cross-linker and ODN, the concentration of ODN and antibody in purified samples was quantified and the molar ratio of ODN to antibody was calculated (S1 Table). Furthermore, to confirm conjugation and characterize the conjugates in terms of structure and molecular weight, we performed size exclusion (SEC)-HPLC and SDS-PAGE analysis.

SEC-HPLC analysis of Trastuzumab showed a symmetrical peak corresponding to monodisperse antibody, a peak with shorter retention time indicative of aggregates and a small peak with longer retention time, likely representing antibody fragments (Fig 1A). The relative area under the curve calculated for each of the peaks was 3.5% for aggregates, 95.4% for monodisperse antibody and 1.1% for fragments. When Trastuzumab-MCC-GpC conjugate was

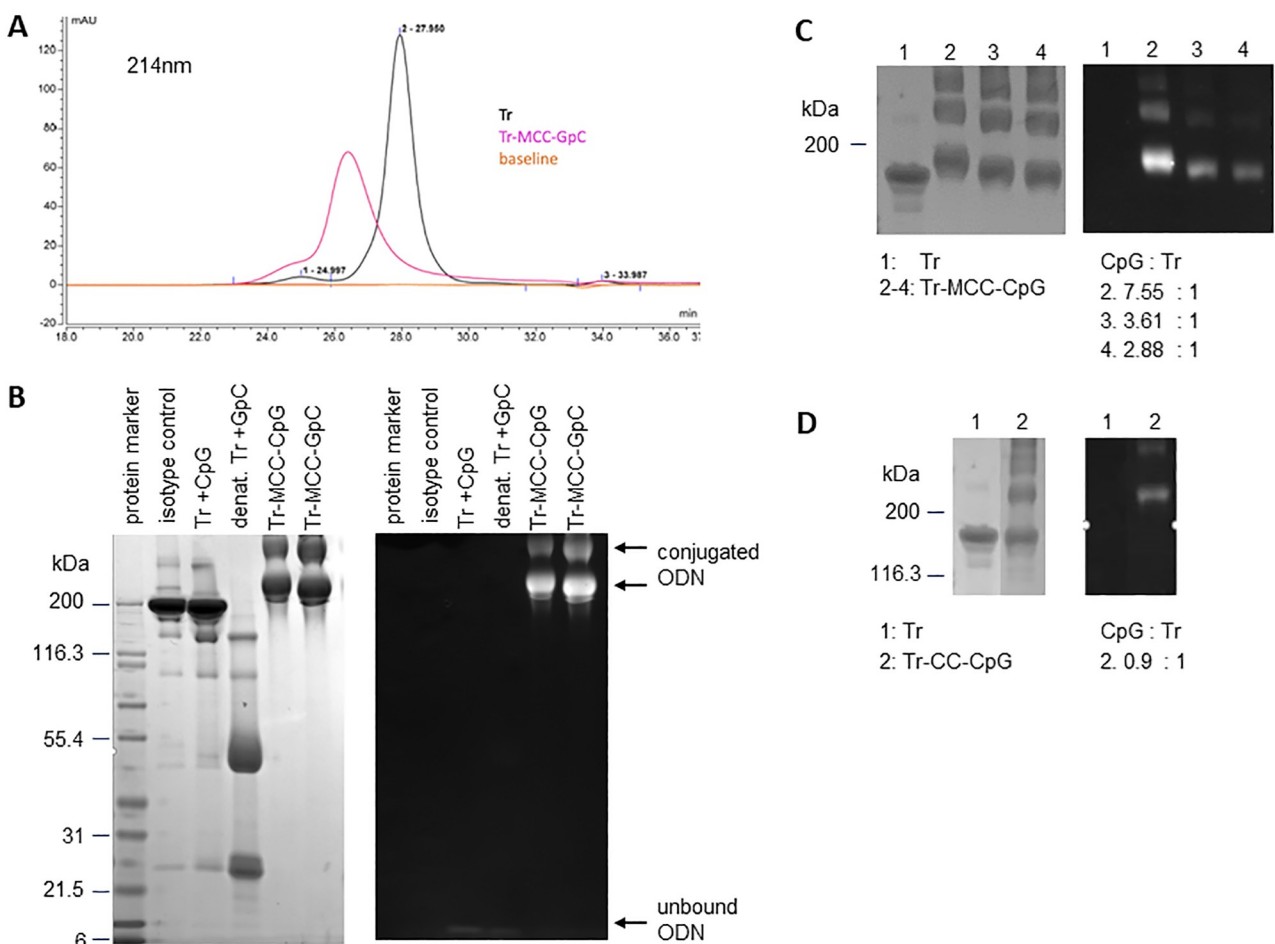

**Fig 1. Characterization of antibody-ODN conjugates by SEC-HPLC and SDS-PAGE.** (A) HPLC analysis of Trastuzumab (Tr) and Trastuzumab-MCC-GpC (Tr-MCC-GpC) conjugate eluted on a TSK G4000SWXL column. The ODN to antibody ratio of the purified conjugate was 6.28. Overlaid chromatograms from one representative analysis are shown. (B) SDS-PAGE analysis of Trastuzumab with unconjugated CpG (Tr +CpG) and Trastuzumab-MCC-ODN. The Tr-MCC-GpC conjugates corresponds to the sample shown in A. The Tr-MCC-CpG conjugate had a ODN: antibody ratio of 9.8. Denatured Tr plus unconjugated ODN was analysed as control as was an IgG1 isotype control Ab. The gel was stained sequentially with Ethidium Bromide (right panel) for visualisation of ODN content and SimplyBlue (left panel) for analysis of protein content. (C) SDS-PAGE analysis of Trastuzumab (Tr) and Tr-MCC-CpG conjugate under non-reducing conditions. Tr-MCC-CpG conjugate (2–4) was generated in PBS buffer with a molar excess of 34 SMCC and 10, 6 and 3 ODN, respectively. The resulting conjugates had a ODN: antibody ratio of 7.55, 3.61 and 2.88:1, respectively. The data are representative of five independent experiments. (D) SDS-PAGE analysis of Tr and Tr-CC-CpG conjugate under non-reducing conditions. The conjugate had an ODN: antibody ratio of 0.9:1 and the analysis is from a single experiment.

analyzed by HPLC, the peak corresponding to monodisperse conjugate eluted faster than the corresponding peak for unconjugated Trastuzumab as expected for a larger molecule. The elution peak of Trastuzumab-MCC-CpG conjugate was smaller and wider than the elution peak of Trastuzumab and slightly asymmetric which is consistent with the expected heterogeneic make-up of SMCC-antibody conjugates. Monodisperse conjugate made up approximately 90% of the sample while higher molecular species represented roughly 9% of the analyte.

In addition, Trastuzumab and preparations of antibody-MCC-ODN conjugates were analyzed by SDS-PAGE. By staining the same gels sequentially for nucleic acid and protein, the co-migration of ODN and IgG was visualized confirming successful conjugation (Fig 1B). As expected, the strength of the ethidium bromide staining correlated with the payload: antibody ratio of conjugates (Fig 1C) whereas unconjugated antibody did not show staining for nucleic

acids (Fig 1B–1D). Staining for protein confirmed that all antibody-ODN conjugate preparations contained higher molecular weight species in addition to monomeric conjugate.

As an alternative to conjugation with SMCC, we explored conjugate generation with the heterobifunctional cross-linker cyanuric chloride or 2,4,6-trichloro-1,3,5,-triazine (CC) which has been used for generation of biotinylated antibodies [16]. The first reactive chlorine of CC can react with the hydroxyl groups of ODN, followed by conjugation of the second reactive chlorine to primary amines on the antibody. We were interested in exploring the use of this alternative cross-linker since it could enable the generation of antibody conjugates with nucleic acid TLR agonists that lack thiol modifications.

For conjugations with CC, the molar ratio of ODN: antibody of purified Trastuzumab-CC-CpG conjugate was 1.2. Any attempt to increase the payload to antibody ratio (PAR) for these CC conjugates failed. In line with the low ratio of ODN: antibody, the conjugate showed weak positive staining with ethidium bromide upon SDS-PAGE analysis (Fig 1D). Furthermore, a portion of the antibody migrated at the same level with the unconjugated control antibody indicating that only part of the sample was successfully conjugated to CpG. Similar to conjugates generated with SMCC, the CC conjugate showed multiple bands migrating slower on the gel indicating the presence of a heterogenous mix of higher molecular weight species.

### *In vitro* characterization of antibody-ODN conjugates

To investigate whether conjugation of CpG ODN to Trastuzumab affected its stimulatory activity, we performed *in vitro* activation assays with murine bone marrow-derived DC (BMDC) generated in the presence of GM-CSF. These GM-CSF BMDC were stimulated overnight with doses of 0.25 and 0.5μg/ml CpG ODN in form of free or conjugated ODN, normalizing samples for ODN content. Similarly, the dose of free Trastuzumab was normalized to Trastuzumab content in the tested Trastuzumab-MCC-CpG conjugate. Upon overnight stimulation, IL-12p40 and IL-6 in the supernatant of the cultures was measured by ELISA. As expected, neither Trastuzumab nor Trastuzumab-MCC-GpC conjugate induced cytokine production by GM-CSF BMDC (Fig 2A and 2B). In contrast, Trastuzumab-MCC-CpG induced levels of IL-6 and IL-12p40 comparable to the equivalent doses of free stimulatory CpG ODN added to GM-CSF BMDC alone or in combination with unconjugated antibody.

The same cytokine induction pattern was observed when Trastuzumab-MCC-CpG conjugate was pre-incubated with HER2-expressing B14.3 HER2 tumor cells before addition of GM-CSF BMDC to the cultures (Fig 2C and 2D). Interestingly, IL-6 levels in the supernatants were approximately two times higher when GM-CSF BMDC were stimulated alone compared to the tumor cell co-cultures. In contrast, the IL-12p40 response of GM-CSF BMDC to conjugated and unconjugated CpG ODN was unaffected by the presence of tumor cells. The most likely explanation for the reduced levels of IL-6 in the presence of tumor cells is increased consumption of the cytokine. However, other more specific mechanisms involving cross-talk between BMDC and tumor cells cannot be excluded [17]. In summary, these results demonstrate that antibody-MCC-CpG conjugates maintain the full immunostimulatory activity of the CpG ODN adjuvant.

Next, we set out to investigate the biological activity of conjugated Trastuzumab. This was crucial since it has been reported that conjugation reactions can impair antibody function including antigen binding and Fc-mediated function [18, 19]. The therapeutic activity of Trastuzumab is partly determined by its ability to inhibit the growth of human HER2 over-expressing tumors. Inhibition of proliferation is dependent on human HER2 binding, which prevents activation of downstream signalling pathways sustaining proliferation [9]. This can be evaluated *in vitro*, by assessing proliferation of HER2 over-expressing tumor cells. Treatment with

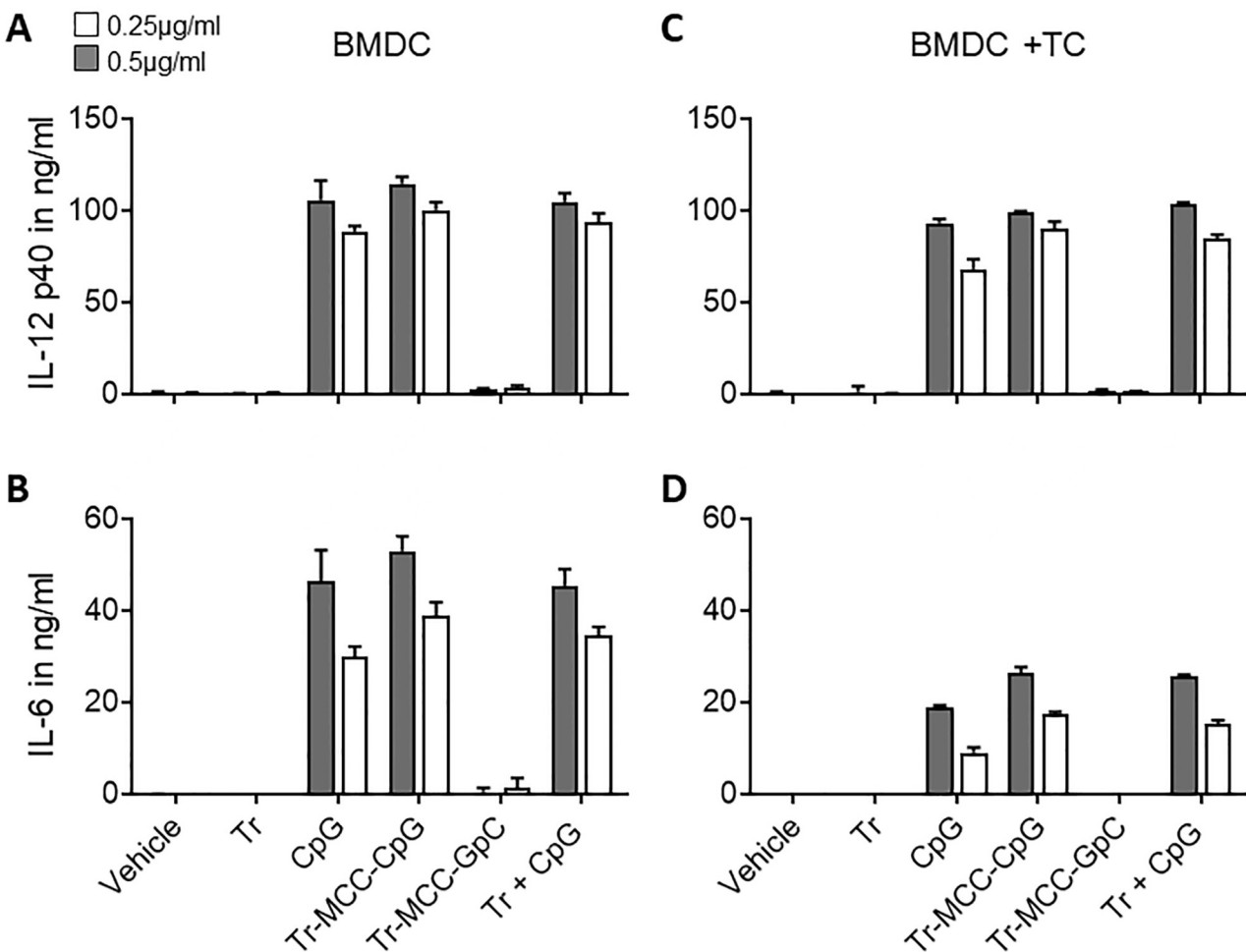

**Fig 2. The stimulatory activity of conjugated CpG ODN was evaluated *in vitro* in DC activation assays.** $0.2x10^5$ GM-CSF BMDC were stimulated overnight alone (A, B) or in the presence of $4x10^5$ B14.3 HER2 tumor cells (C, D) with 0.25ug/ml (grey bars) and 0.5ug/ml (white bars) of ODN, either free (stimulatory CpG ODN) or conjugated (stimulatory CpG and non-stimulatory GpC ODN) with SMCC to Trastuzumab. Untreated samples (Vehicle) Trastuzumab and Trastuzumab-MCC-GpC conjugates were used as negative control. A combination of unconjugated Trastuzumab and CpG ODN (Tr +CpG) was also tested. IL-12p40 (A, C) and IL-6 (B, D) in the supernatants of overnight cultures was measured by sandwich ELISA. Data are mean ± SD of three technical replicates and is representative of three independent experiments with different batches of conjugates.

unconjugated Trastuzumab inhibited proliferation of the human HER2 over-expressing tumor cell lines BT474 and OE-19, but not of the triple-negative MDA-MB-231 breast cancer cell line used as HER2-negative control (Fig 3A–3C). Trastuzumab-MCC-CpG conjugate was less efficient in inhibiting proliferation of the HER2 over-expressing cell lines than the unconjugated Trastuzumab and showed a shift in the dose-response curve (Fig 3A and 3B). The $Log_{10}IC_{50}$ for Trastuzumab-MCC-CpG conjugate was 0.27 for BT474 cells and 0.43 for OE19 cells compared to a $log_{10}IC_{50}$ for Trastuzumab of 0.18 and -0.09, respectively. These results suggested that the tumor growth inhibitory activity of Trastuzumab is partially impaired upon conjugation.

The immune system contributes to the therapeutic effects of Trastuzumab through mechanisms such as antibody-dependent cell-mediated cytotoxicity (ADCC) and antibody-dependent cell-mediated phagocytosis (ADCP) [20]. To evaluate the impact of CpG conjugation on these anti-tumor activities mediated via the antibody Fc region, we performed an ADCC assay

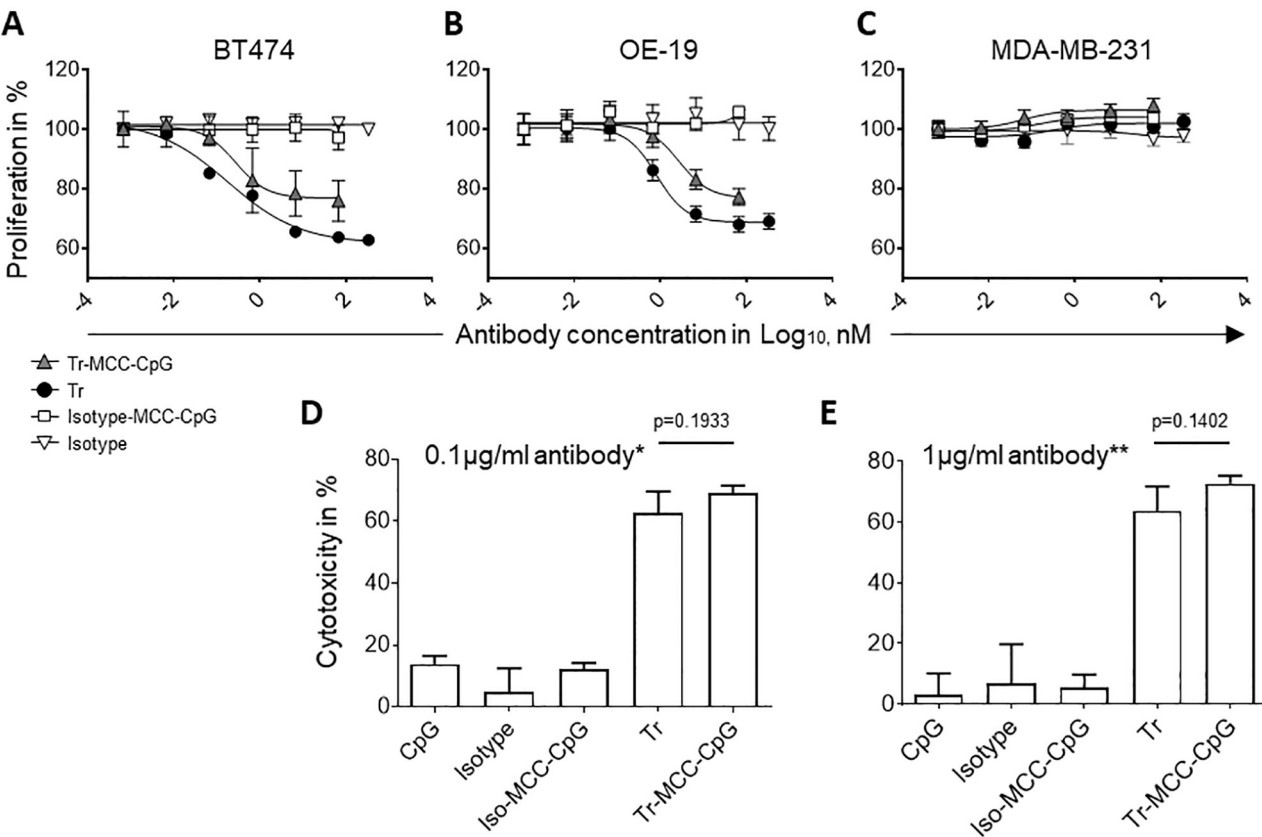

**Fig 3. In vitro evaluation of the tumor growth inhibitory and ADCC activity of Trastuzumab-ODN conjugate.** HER2 over-expressing Trastuzumab-sensitive human cancer cell lines BT474 and OE-19 (A, B) and the HER2-negative/low cell line MDA-MB-231 (C) were incubated with increasing concentrations of Trastuzumab (black dots), Trastuzumab-MCC-CpG (grey triangles), isotype (white triangles) or isotype-MCC-CpG (white squares). Trastuzumab-MCC-CpG conjugate had a ODN: antibody ratio of ~1.7:1. After 48h MTS substrate was added to the wells and signal was measured after 30 min incubation. Data was expressed as percent of untreated controls. Dose-response curves generated with non-linear fit equation with variable slope in GraphPad. Data are means ± SEM of three technical replicates and is representative of two independent experiments. (D, E) The ADCC function of conjugates was evaluated in a calcein-release assay with BT474 target cells incubated with 0.1µg/ml (D) or 1µg/ml (E) antibody or conjugate corresponding to -1.171 and 0.1171 Log10 nM of conjugate, respectively. Concentrations of conjugated CpG were 31.75ng/ml (D) and 317.5ng/ml (E), respectively. Equivalent concentrations were used for free CpG ODN and Iso-MCC-CpG conjugate. PBMCs from three healthy donors were added at a effector: target cell ratio of 1:25 and the co-culture was incubated for 16h. Afterwards, calcein released in the media was measured. Means ± SD of data generated with PBMCs from three healthy donors collected in two independent experiments. For the experiments, different batches of Trastuzumab-MCC-CpG with ODN: antibody ratios of 1.8–2: 1 were used. Student's t test was used for statistical analysis.

measuring calcein release by human PBMC effector cells upon encounter of BT474 target cells. As expected, isotype control antibody or CpG ODN, either free or conjugated to the isotype control antibody only induced minimal levels of cytotoxicity in this assay (Fig 3D and 3E). Surprisingly, the ADCC activity of Trastuzumab-MCC-CpG conjugate was comparable to that of unconjugated Trastuzumab despite the decrease in its tumor cell growth inhibitory activity. This suggests that the functionality of the antibody Fc region is not affected by the conjugation to ODN. However, we cannot rule out an enhancing effect of the conjugated CpG ODN on ADCC in our assay compensating for a decrease in the binding ability of the antibody. CpG ODN have been implemented in enhancing ADCC in other studies [21–23]. The ADCC enhancement by TLR9 agonists is thought to be mediated by an indirect mechanism that involves IFN-α and IL-12 produced by DC [24, 25]. Since results from *in vitro* ADCC assays are not necessarily predictive for functionality *in vivo*, investigating this aspect in a humanized mouse model could be informative.

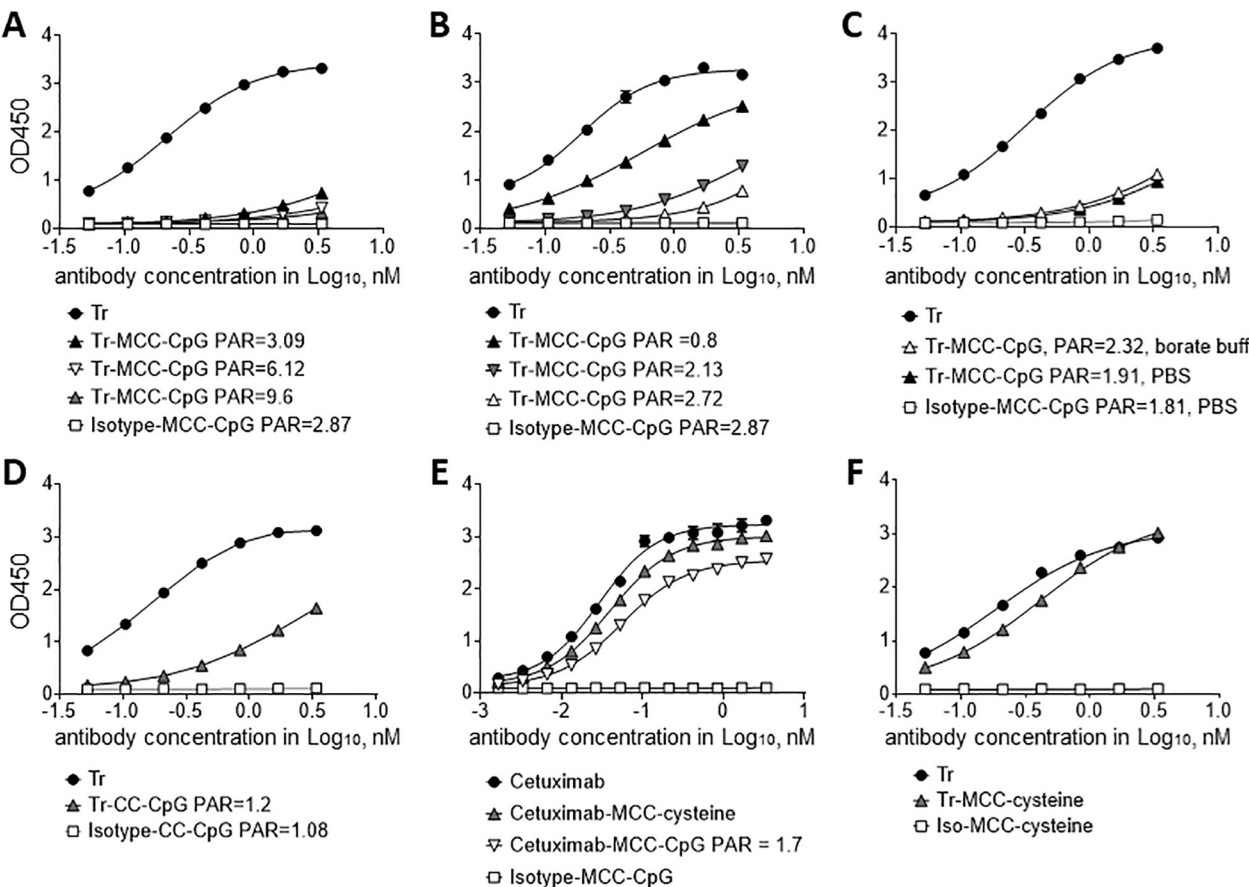

**Fig 4. Impact of conjugation conditions on antigen binding properties of conjugates.** Trastuzumab (A-C) were conjugated in either PBS (A-C) or borate buffer (C) at various conditions of SMCC and CpG molar excess resulting in conjugates with varying payload to antibody ratios (PAR) as indicated. (D) Trastuzumab was conjugated to CpG using CC. (E) Cetuximab was conjugated to cysteine or CpG using SMCC. (F) Trastuzumab was conjugated to cysteine using SMCC. (A-F) Isotype control antibody conjugated to CpG or cysteine are shown as negative controls. Resulting conjugates were analyzed by ELISA using immobilised human HER2 (A-D, F) or EGFR (E) and anti-human IgG detection antibody. Binding curves were generated using non-linear regression (variable slope) function in GraphPad Prism. Data are representative of at least two independent experiments.

We next evaluated the antigen binding properties of the conjugates using an ELISA assay with immobilized human HER2. In this assay, Trastuzumab bound HER2 in a dose dependent manner (Fig 4). Isotype-MCC-CpG conjugate did not bind to HER2, indicating that CpG does not interfere with the assay. All tested Trastuzumab-ODN conjugates showed markedly altered binding to HER2 compared with the unconjugated antibodies. There is a negative correlation between the number of CpG molecules incorporated into the conjugates and the binding affinity of the conjugates to HER2 for all Trastuzumab-ODN conjugates (Fig 4A and 4B). Conjugate with a CpG to Trastuzumab ratio of <1 showed strikingly improved binding over conjugates with a CpG to antibody ratio >2 (Fig 4B). Conjugate generation in borate buffer compared to PBS did not improve the antigen binding of the generated conjugate (Fig 4C). Likewise, when binding of CC conjugates to HER2 was assessed by ELISA reduced binding of Trastuzumab-CC-ODN conjugate was observed in comparison to unconjugated Trastuzumab (Fig 4D).

We concluded from these binding experiments, that the loss in antigen binding activity is most likely a consequence of steric hindrance by ODN molecules that seem to preferentially be

conjugated to a site close to the antigen-binding site of the Trastuzumab antibody. This phenomenon is antibody specific and depends on the location of reactive groups within the amino acid sequence of the antibody and was not observed for similar antibody-ODN conjugates generated for DC targeting [6]. Also, the binding of Cetuximab was only slightly affected upon conjugation to CpG (Fig 4E). When cysteine instead of ODN was conjugated to Trastuzumab using the same conjugation protocol, no loss in antigen binding activity was detected despite a cysteine to antibody ratio of ~3:1 (Fig 4F). The smaller molecular weight of cysteine with 121.16Da does not lead to steric hindrance of the antigen binding site upon conjugation in contrast to CpG ODN which has a molecular weight of 6,345Da.

## Site-specific conjugate generation and *in vitro* characterization of ThioMab conjugates

Since the conventional stochastic Trastuzumab conjugates generated with SMCC and CC showed significantly altered antigen binding activity, we decided to explore a site-specific conjugation strategy. In addition to unaltered antigen binding properties, site-specific conjugates promise higher homogeneity and *in vivo* stability, influencing both safety and therapeutic efficacy of conjugates [26]. Among site-specific conjugation methods, we chose to use ThioMab technology, which involves genetically engineering cysteines at specific sites in the structure of Trastuzumab and specifically conjugating our payload of choice via a maleimide-containing linker to the thiol groups of these cysteines [27].

It has been shown that SMCC-mediated conjugation of modified Trastuzumab in which valine has been exchanged with cysteine at position 205 in the light immunoglobulin (Ig) chain (LC-V205C) yields ADC with superior *in vivo* stability and enhanced therapeutic function [28]. This strategy avoids steric hindrance of the payload, which is conjugated at a distance from the antigen-binding site of the antibody.

We engineered cysteines at LC-V205C in the Trastuzumab construct and generated Trastuzumab ThioMab (TH-Trastuzumab). Purified TH-Trastuzumab migrated on a SDS PAGE gel similarly to the parent antibody confirming that both recombinant antibodies had the same molecular weight (Fig 5A). For conjugation of CpG ODN to TH-Trastuzumab, we optimized a previously published protocol for generation of ADC [29]. Efficient reduction and reoxidation of TH-Trastuzumab was verified by SDS-PAGE analysis (Fig 5A). The ODN to antibody ratio of the generated TH-Trastuzumab-CpG conjugates were consistently in the 1.5–1.9 range. In contrast to stochastic Trastuzumab-MCC-CpG conjugates (Fig 2B), TH-Trastuzumab-MCC-CpG conjugates only showed minimal multimer content as expected for a site-specific conjugation approach (Fig 5B).

To evaluate the biological activity of TH-Trastuzumab-MCC-CpG conjugate, we performed a DC activation assay with BMDC. In comparison to free CpG ODN, conjugated CpG ODN induced production of comparable levels of IL-12p40 and IL-6 confirming that the conjugation doesn't alter the stimulatory activity of the ODN (Fig 5C). Next, we assessed the functionality TH-Trastuzumab-MCC-CpG conjugate to inhibit the proliferation of HER2-sensitive tumor cell lines in the MTS assay. The decrease in the proliferation of human Trastuzumab-sensitive cells was comparable between TH-Trastuzumab-MCC-CpG conjugate and free TH-Trastuzumab (Fig 5D). The $Log_{10}IC_{50}$ values for these two treatment groups were comparable for BT474 cells and OE19 cells. As expected, antigen binding of TH-Trastuzumab-MCC-CpG conjugate was comparable to antigen binding of unconjugated Trastuzumab (Fig 5E).

Collectively, the *in vitro* analysis of the TH-Trastuzumab-MCC-CpG conjugate confirmed that its biological activity in terms of antigen binding and tumor cell growth inhibitory activity

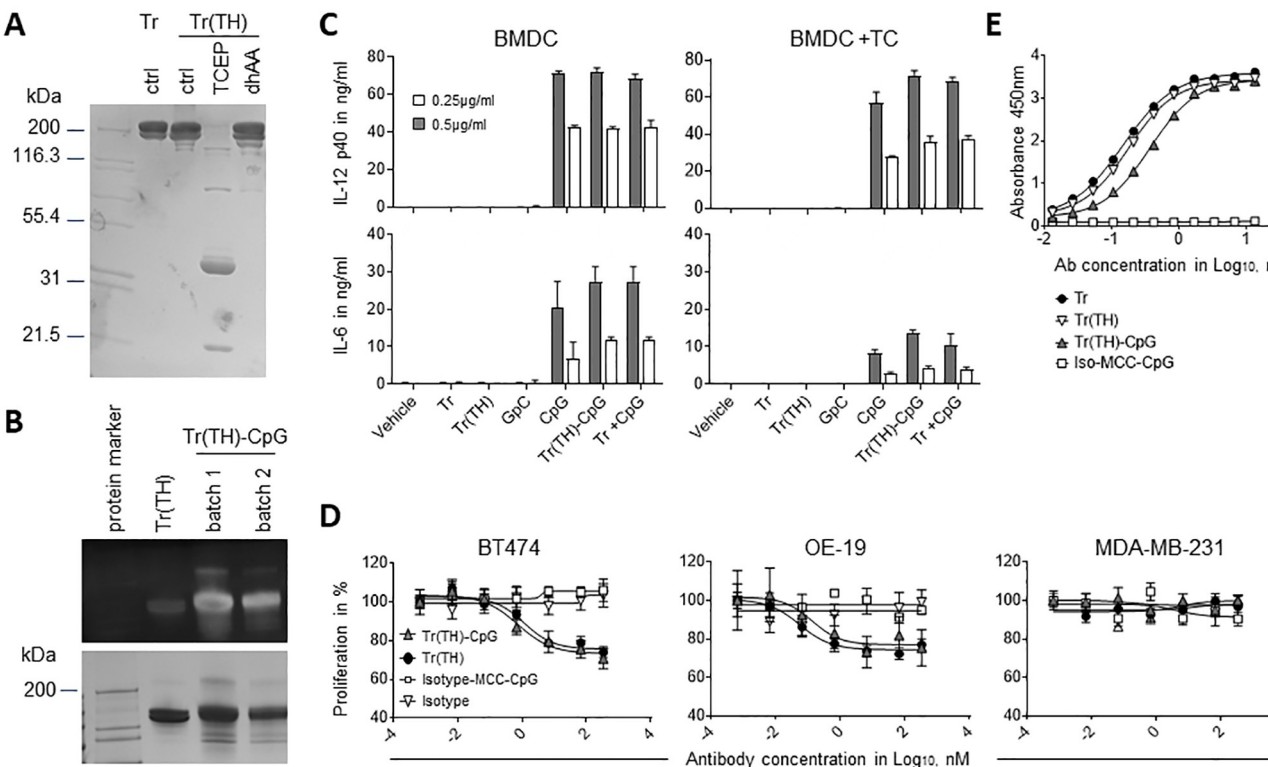

**Fig 5. Generation and *in vitro* evaluation of TH-Trastuzumab-MCC-ODN conjugate.** (A) SDS-PAGE analysis of TH-Trastuzumab in native, reduced and reoxidated form on 4–20% gel stained with SimplyBlue. (B) SDS-PAGE analysis of TH-Trastuzumab-MCC-CpG conjugates under non-reducing conditions. The gel was sequentially stained with SYBR Gold (top) to visualise the ODN content, then with SimplyBlue (bottom) for protein content. The Data are representative of three independent experiments. (C) The immunostimulatory activity of TH-Trastuzumab-MCC-CpG was assessed *in vitro* in a BMDC activation assay. $1\times10^5$ GM-CSF derived BMDC were stimulated overnight with free or conjugated CpG in the presence (BMDC +TC) or absence (BMDC) of $4\times10^5$ B14.3 HER2 tumor cells. IL-12p40 and IL-6 levels in the culture supernatants were determined by sandwich ELISA. Samples were normalised for concentration of CpG of 0.5μg/ml or 0.25μg/ml. Data are mean ± SD of three technical replicates and is representative of two independent experiments. (D) The anti-proliferative effect of TH-Trastuzumab-MCC-CpG was assessed in the MTS assay. Trastuzumab-sensitive human tumor cell lines BT474 and OE19, and the HER2-negative/low Trastuzumab-resistant control cell line MDA-MB-231 were cultured for 48 h with antibody or antibody-MCC-CpG conjugate. The antibody: ODN ration of the conjugates was approximately 1.6:1. The data were normalised to the proliferation of untreated control cells and expressed as percent proliferation. Data are representative of two independent experiments with individual batches of conjugates. (E) The antigen binding activity of TH-Trastuzumab-MCC-CpG was compared to unconjugated Trastuzumab in the ELISA with immobilised recombinant human HER2. The ODN: antibody ratio of the conjugates was approximately 1.7:1. The data are representative of four independent experiments with individual batches of conjugates. dhAA: haloalkane dehalogenase; TCEP: Tris (2-carboxyethyl) phosphine.

compared much more favourably to the activity of unconjugated Trastuzumab than that of the various stochastic Trastuzumab-MCC-CpG conjugates that we tested. We therefore proceeded with the TH-Trastuzumab-MCC-CpG conjugate for evaluation of its therapeutic functionality in an *in vivo* mouse model.

## Evaluation of the therapeutic activity of TH-Trastuzumab-MCC-CpG conjugate

For investigation of Trastuzumab-mediated delivery of adjuvants to the tumor tissue *in vivo* we elected to work with a modified version of the B16 melanoma pseudo-metastasis model which we had used for past studies [6, 14]. In order to accommodate the use of Trastuzumab in this model, we transduced B14.3 cells with a lentiviral vector encoding human HER2. We chose this transgenic mouse model for specifically studying the targeted approach of antibody-

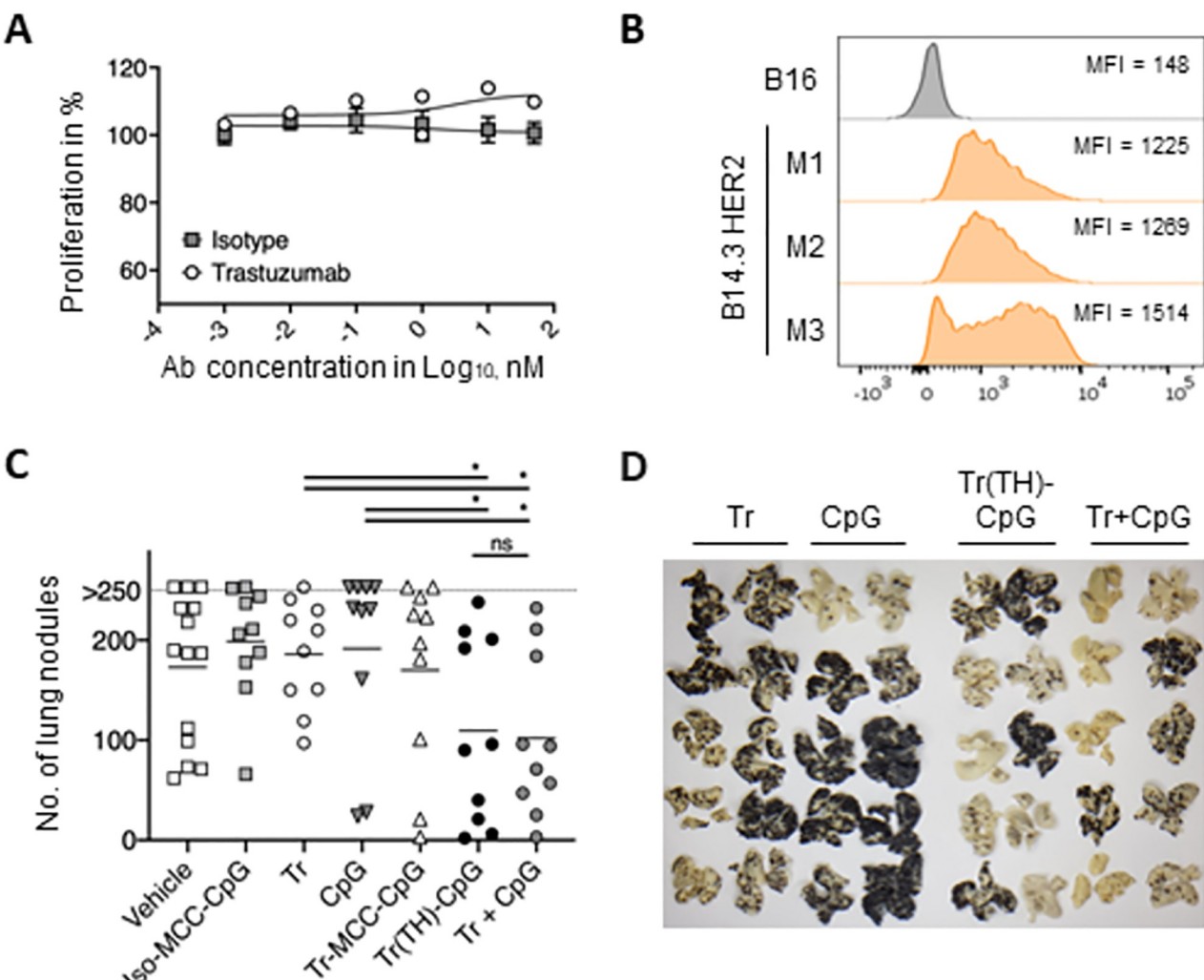

**Fig 6. Anti-tumor response upon repeated treatment with TH-Trastuzumab-MCC-CpG in the B14.3 HER2 pseudo-metastasis mouse model.** (A) B14.3 HER2 cells were cultured in the presence of Trastuzumab for 48h and tumor cell proliferation was measured in the MTS assay. (B) Expression of human HER2 was determined by flow cytometry on B14.3 HER2 cells retrieved from 3 representative mouse lungs 15 days after tumor inoculation. B16 tumor cells served as HER2-negative control. The MFI of each samples is depicted. (C) Mice (n = 10 for all groups, except vehicle n = 9) were inoculated with B14.3 HER2 cells on day 0 and treated at days 4, 8 and 11 with 2mg/kg body weight of antibody or antibody conjugate. Mice were treated with Trastuzumab (Tr) and CpG ODN (CpG) alone or in combination. The dose of CpG provided in free form was equivalent to the dose of conjugated CpG. Trastuzumab-MCC-CpG (Tr-MCC-CpG) and TH-Trastuzumab-MCC-CpG (Tr(TH)-CpG) conjugates were compared and isotype control antibody-MCC-CpG (iso-MCC-CpG) conjugates were used as control. The number of pulmonary nodules on the lung surface was determined at day 18 post-tumor inoculation. Results for individual mice and mean number of lung tumor nodules per group are shown. Mann-Whitney T test; * p<0.05; ns—not significant. (D) Representative images of mouse lungs bearing B14.3 HER2 tumors at day 18 post tumor inoculation are shown.

mediated delivery of TLR9 agonist to the tumor tissue and the induction of anti-tumor immunity as a consequence of local innate immune activation.

B14.3 cells were lentivirally transduced to express human HER2, (S1 Fig). The proliferation of the B14.3 HER2 tumor cells was not directly affected by treatment with Trastuzumab (Fig 6A), highlighting that any change of tumor growth that would be observed *in vivo* would come from an indirect effect. It is also worth noting that the parental B16 cell line does not express TLR9 (S1 Fig) and that upon inoculation of mice, B14.3 HER2 tumor cells maintain a

moderate degree of human HER2 expression (Fig 6B) but show a decrease in HER2 expression compared to cultured cells (S1 Fig).

To investigate the therapeutic effect of the conjugates, mice were treated with a dose of 2mg/kg body weight of Trastuzumab or TH-Trastuzumab-MCC-CpG conjugate translating approximately into 40μg of antibody per mouse. This dose is in the range of clinical used doses and preclinical mouse models demonstrated successful accumulation of Trastuzumab in HER2-expressing tumors [30–32]. At an ODN to antibody ratio of 1.8–2:1, 40μg of TH-Trastuzumab-MCC-CpG carried roughly 2.5μg of conjugated CpG. The different treatment groups were normalized to equal the antibody or the ODN dose as appropriate. Therapeutic treatment was initiated on day 4 post tumor-inoculation and repeated twice, on day 8 and day 11 (S2 Fig) emulating published studies showing CpG efficacy in tumor immunotherapy [33, 34]. Tumor-bearing mice did not respond to treatment with Trastuzumab or CpG alone (Fig 6C). In contrast, treatment with a combination of unconjugated Trastuzumab and CpG ODN showed reduced tumor growth in terms of lung nodule count (Fig 6C) indicating to a synergistic effect of both treatments. No reduction in number of lung nodules was observed for mice treated with isotype control antibody-CpG conjugates (Fig 6C). Mice treated with stochastic Trastuzumab-MCC-CpG conjugates showed no significant decrease in tumor burden compared to mice treated with equimolar doses of either Trastuzumab or CpG ODN (Fig 6C). In contrast, mice treated with TH-Trastuzumab-MCC-CpG conjugates had significantly lower numbers of lung tumor nodules at day 18 (Fig 6C and 6D) showing that site-specific conjugation generates functional conjugate whereas the biological activity of stochastic Trastuzumab-TLR9 agonist conjugate is impaired. However, the observed anti-tumor activity of TH-Trastuzumab-MCC-CpG treatment group was similar to Trastuzumab plus CpG treated mice. While this could be interpreted as proof that non-targeted delivery of CpG ODN is as efficient as conjugate-mediated targeted delivery, it is important to point out that intravenous administration of the therapeutic favors delivery to the lung. Therefore, it will be important to examine alternative treatment routes and/or tumor models in the future to specifically investigate the efficacy of conjugate-mediated targeted delivery to distant tumor sites.

We investigated the pharmacokinetics of the TH-Trastuzumab-MCC-CpG conjugates by monitoring the presence of human IgG1 antibody in the serum of tumor-bearing mice at various time points after three doses of treatment. The serum concentration of Trastuzumab or conjugates was highly variable within groups (S2 Fig). Co-injection of CpG ODN with Trastuzumab did not alter the serum concentration of the antibody. The stochastic Trastuzumab-MCC-CpG conjugate showed a significantly lower serum concentration on both days. In contrast, the serum concentration of TH-Trastuzumab-MCC-CpG was only slightly lower than that of Trastuzumab at both time-points. The faster clearance of stochastic Trastuzumab-MCC-CpG conjugate may be a contributing factor to its failure in inducing a therapeutic effect.

Collectively, the data indicate that treatment with site-specific TH-Trastuzumab-MCC-CpG conjugate, but not stochastic Trastuzumab-MCC-CpG conjugate delays tumor growth in mice with B14.3 HER2 pulmonary tumors which correlated with the biological activities of these conjugates in the *in vitro* HER2 binding assay.

## Characterization of the TH-Trastuzumab-MCC-CpG-induced T cell response

One of the major effects of CpG-mediated TLR9 signalling is the induction of adaptive immune responses with a Th1 phenotype and CTL effector functions [35]. To evaluate the expansion of T cells in response to TH-Trastuzumab-MCC-CpG conjugate treatment *ex vivo*,

spleens were harvested at day 18 of the experiment and splenocytes were analyzed by flow cytometry for expression of the T cell activation markers CD44 and CD62L (S3 Fig). Mice showed comparable numbers of total CD4 and CD8 T cells across treatment groups (S4 Fig). Most CD44$^+$ T cells were also CD62L$^+$, indicating a central memory phenotype rather than a CD62L-negative effector memory phenotype (S4 Fig). Percentages of CD44$^+$ CD62L$^+$ cells in the CD8 and CD4 T population were significantly higher in mice treated with CpG compared to mice treated with vehicle or Trastuzumab, and this was independent of the therapeutic format of CpG which was provided alone, in combination with Trastuzumab or in form of conjugate (Fig 7A). The same trend was observed for absolute counts of CD44$^+$ CD62L$^+$ CD8 T cells, but the differences in absolute counts was not significant for mice treated with Trastuzumab-MCC-CpG or Trastuzumab in combination with CpG ODN compared to mice treated with Trastuzumab (Fig 7A). However, a statistically significant increase in absolute counts of CD44$^+$ CD62L$^+$ cells was observed for CD4 T cells in all groups receiving treatment with free or conjugated CpG (Fig 7A).

Next, we investigated the antigen-specific T cell response. Splenocytes from mice receiving three treatment doses were labelled with CFSE and cocultured for three days with DC pulsed with a cocktail of recombinant human HER2 plus peptides specific for antigens expressed by B14.3 HER2 melanoma cells. The percentage of proliferating T cells was determined by flow cytometry gating on CD44$^+$ activated T cells (S3 Fig). Proliferation of restimulated cells from tumor-bearing vehicle-treated mice was 31.3% ± 5.9 and 17.5% ± 3.1 for CD4 and CD8 T cells, respectively (Fig 7B). Treatment with TH-Trastuzumab-MCC-CpG, but not Trastuzumab-MCC-CpG or isotype-MCC-CpG induced superior proliferation of antigen specific CD8 T cells compared to the vehicle control. However, while the proliferation of antigen-specific CD8 T cells from mice treated with TH-Trastuzumab-MCC-CpG was greater than for CpG-treated mice, there was no significant increase in comparison to mice treated with Tr or Tr plus CpG. In contrast, for CD4 T cells TH-Trastuzumab-MCC-CpG treated mice showed increased antigen-specific proliferation in comparison to mice treated with unconjugated Trastuzumab, unconjugated CpG or a combination of the two. These results indicate that treatment with TH-Trastuzumab-MCC-CpG conjugate is superior in inducing antigen-specific CD4 T cells in comparison to a combination of unconjugated Trastuzumab and CpG, even though both treatment groups are similarly effective in reducing tumor burden. This suggests that distinct mechanisms may be responsible for the tumor growth reduction observed in mice treated with a combination of Trastuzumab and CpG in conjugated versus unconjugated form.

## Discussion

Our comparative evaluation of the antigen-specific binding activity, the tumor growth inhibitory capacity and the therapeutic anti-tumor efficacy of various Trastuzmab-CpG conjugates demonstrated conclusively that site-specific conjugation chemistry is superior to conventional stochastic conjugation protocols. However, successful stochastic conjugation of Trastuzumab with the cross-linker SMCC with various payloads was reported by other groups and was employed for the generation of the approved ADC Trastuzumab-emtansine [18, 36, 37]. An important factor that determines whether this conjugation approach is successful for Trastuzumab is the size of the payload. This effect has been widely observed and was demonstrated in our study by the conjugation of cysteine, which did not impair the functionality of the Trastuzumab antigen-binding site whereas larger payloads such as CpG ODN led to steric hindrance [38, 39]. It is important to note that the obstruction of the antigen-binding site by large payloads is dependent on the location of accessible SMCC-reactive groups on the antibody and is thereby sequence-specific. There are 88 lysine groups in Trastuzumab which represent

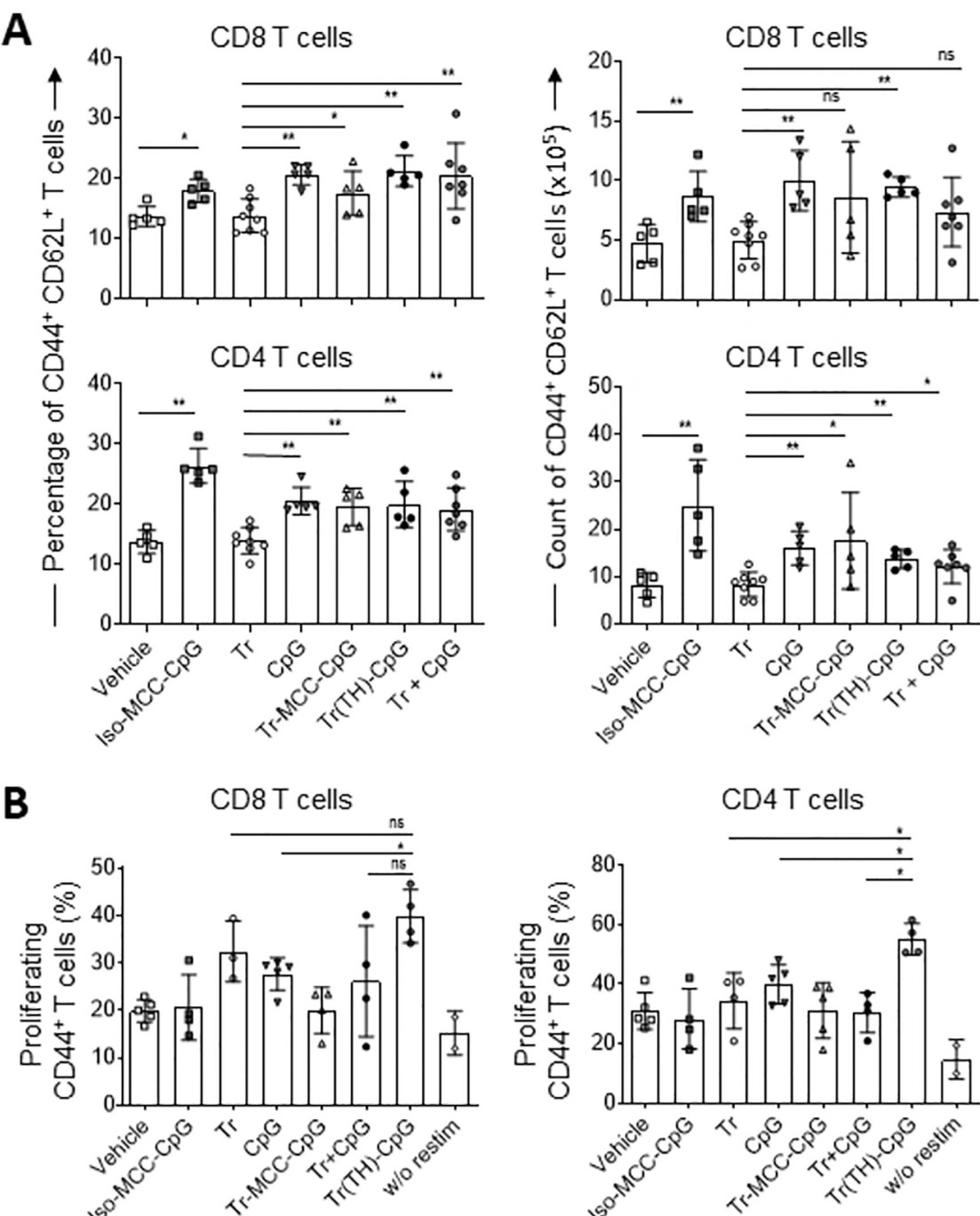

**Fig 7. Cellular immune responses in mice upon repeated treatment with TH-Trastuzumab-MCC-CpG.** Mice were treated on days 4, 8 and 11 post tumor inoculation with 2mg/kg body weight of the indicated antibodies or antibody conjugates or with a dose of free CpG equivalent to the CpG dose applied in conjugated form. Splenocytes were isolated on day 18. (A) Splenocytes were stained and analyzed *ex vivo* by flow cytometry to determine the percentage and the absolute counts of activated CD44+ CD62L+ central memory CD4+ and CD8+ T cells in the samples. Each symbol represents a mouse treated with vehicle (n = 5), Iso-MCC-CpG, Tr-MCC-CpG, Tr(TH)-CpG or Tr (n = 8 for each), and Tr + CpG (n = 7). Means ± SD of samples from two independent experiments are depicted. Data were analyzed by Mann-Whitney two tailed test; *p<0.05; **p<0.01; ns—not significant. (B) Splenocytes were labelled with CFSE and incubated with an antigen mix and antigen-presenting cells for 72h. The cells were harvested and the percentage of proliferating CD44+ CD4+ and CD8+ T cells was determined by flow cytometry. Each data point represents an individual mouse (n = 4 or 5). Bars show mean values ± SD. Mann-Whitney two tailed tests; *p<0.05; ns—not significant. Data are from one experiment.

available solvent NH2 groups amenable to SMCC-mediated conjugation which give rise to highly heterogenous populations of conjugates [37, 40]. This explains why we did not observe a decrease in the functionality of a DEC205-specific antibody-CpG conjugate used in a previous study for targeted delivery of adjuvant to DC [6]. Surprisingly, other studies generating Trastuzumab-CpG conjugates by stochastic conjugation protocols did not report impaired functionality of the antibody [41, 42].

While impairment of the antigen-binding activity may not be an issue for the majority of antibodies, stochastic conjugation methods are regarded as suboptimal since they yield conjugate preparations that are structurally heterogenous, potentially containing antibodies that bear no payload and antibodies that are conjugated to a degree that abolishes their activity [18, 36, 37]. Conjugate heterogeneity has been demonstrated to contribute to poor therapeutic activity [43, 44]. In order to ensure more homogenous conjugate preparations, the use of site-specific conjugation technology of engineered antibodies with prominent reactive site chains at strategically optimal locations within the antibody structure are favored irrespective of the antibody sequence and the payload size. We decided to use ThioMab technology for site specific conjugation of Trastuzumab since it represents an elegant solution amenable to a wide variety of antibodies [7, 28]. The bioassays in our study demonstrated clearly that site-specific TH-Trastuzumab-CpG conjugates fully retain their antigen binding their tumor cell growth inhibitory and their ADCC-inducing activity.

In contrast to the functionality of the antibody, the adjuvant activity of CpG ODN was not affected by the conjugation procedure independent of the cross-linker used or whether conjugation was site-specific or stochastic, as demonstrated in the DC activation assay. While we generated stochastic conjugates with various CpG to antibody ratios from 0.4:1 to 9.16:1, the ratio of CpG to antibody for site-specific conjugates was limited by the introduction of only one engineered cysteine site at position 205 in the antibody light chain resulting in conjugates with 1.5 to 1.9 CpG molecules per antibody molecule. Future studies will have to clarify whether the introduction of additional engineered cysteine moieties and an increase in the payload of site-specific conjugates is of therapeutic benefit.

From the *in vitro* characterization of the conjugates, we concluded that site-specific TH-Trastuzumab-CpG conjugates should be fully functional *in vivo*. Targeting of TLR9 agonist to the tumor tissue is thought to enable the priming of antigen-specific T cells in the draining lymph nodes, to promote the tumor infiltration with effector cells and to disable immune checkpoints in the tumor tissue leading to efficient anti-tumor responses [45]. While alternative approaches such as targeting antigen-presenting cells may be more effective in promoting T cell priming, the use of a therapeutic tumor-targeting antibody such as Trastuzumab has the added potential benefit to boost synergistic effects between the TLR agonist and the therapeutic antibody.

We chose a modification of the B16 melanoma pseudo-metastasis model to investigate the therapeutic efficacy of the conjugates, since C57BL/6 mice are fully immunocompetent while the B16 cancer model is poorly immunogenic, has well-defined tumor-associated antigens and requires strong therapeutic intervention strategies to induce anti-tumor immunity [46]. We transduced a modified B16 cell line with a lentiviral vector expressing human HER2. This strategy allowed us to use Trastuzumab which is established clinically for the treatment of HER2-expressing breast cancer patients rather than working with an equivalent mouse HER2-specific antibody and mouse HER2-positive cancer cell lines. Naturally, a highly engineered mouse model has caveats. The most prominent of these caveats is the fact that human HER2 represents a foreign antigen in mice. The expression of a foreign antigen is the likely reason for a noticeable reduction in tumor burden for B14.3 HER2 cells when compared to the parental HER2-negative cells (S1 Fig) and may contribute to the variability in tumour burden

within all treatment groups. The observation that the engineered B14.3 HER2 cells were resistant to Trastuzumab-mediated growth inhibition *in vitro* meant that any therapeutic benefits of the conjugates observed *in vivo* in the cancer model must be exclusively driven by the immune system. This includes the synergistic effect observed for Trastuzumab and CpG ODN in conjugated or unconjugated form which is observed in the absence of a direct tumor growth inhibiting effect of the therapeutic antibody.

A further caveat of the cancer model was the use of the engineered melanoma cells in a pseudo-metastasis model with the injected tumor cells establishing lung nodules. Upon intravenous administration of the conjugates the first capillary bed reached is in the lung [47]. It is therefore possible, that the anatomic location of the tumors in our mouse model favoured efficient distribution of non-targeted CpG to the lung tissue containing the tumor nodules. This may represent a contributing factor for the absence of a significant difference in the tumor burden between TH-Trastuzumab-CpG conjugate and co-injection of unconjugated antibody and CpG in the tumor model. Subcutaneously implanted human HER2-positive tumor cells such as the Trastuzumab-sensitive BT474 cell line in PBMC-engrafted immunodeficient mice may represent an alternative *in vivo* model that excludes efficient access of untargeted CpG ODN to the tumor tissue. In addition, the presence of human leukocytes would also allow for full engagement of the Fc region of the therapeutic antibody with the Fc receptors on human leukocytes and thereby fully support immune effector mechanisms such as ADCC and APCC. Despite all the advantages of a humanized mouse model, it also has caveats such as the missing crosstalk between stromal cells and the immune system and the lack of fully functioning secondary lymphoid organs.

The difference in tumor burden between Trastuzumab and TH-Trastuzumab-CpG conjugate shows that the conjugated antibody represents a suitable vehicle for co-delivery of the adjuvant and expands the therapeutic activity of the antibody. Furthermore, our results show that TH-Trastuzumab-CpG conjugate is more efficient in inducing antigen-specific T cell responses than an unconjugated combination of the antibody and the stimulatory ODN. This suggests that co-delivery of the adjuvant and the antibody in form of a conjugate promotes a tumor-targeted antigen-specific immune response rather than just elevating systemic undirected innate immune activation. The induction of cellular immunity against cancer cells and the support of cell-mediated cytotoxic effector mechanisms is thought to be a crucial factor in the success of immunotherapeutic approaches. Thus, this finding is important. Induction of antigen-specific immunity against cancer cells may have a bigger impact in cancer models that are less restricted in their duration than the pseudo-metastasis model since differences in tumor eradication may need more time to become visible. Likewise, a higher treatment dose or a prolonged treatment regime may increase differences in treatment efficacy between conjugated and unconjugated co-administration of Trastuzumab and CpG.

Other studies investigating the immunotherapeutic potential of antibody-conjugated TLR agonists have also reported the induction of beneficial anti-tumor responses. The mechanisms mediating the anti-tumor immune response are study-specific and depend on a combination of antibody target, TLR agonist and tumor model with different studies reporting anti-tumor response mediated by T cells, NK cells and/or myeloid cells [21, 48, 49]. In addition, conjugates can be designed to induce apoptosis in tumor cells upon antibody-mediated uptake of TLR3 agonists [50]. Thus, the immune-stimulatory properties of TH-Trastuzumab-CpG conjugates may not be limited to the induction of T cell responses, but other tumor models will be required to investigate potential additional functional mechanisms.

Further aspects of the treatment strategy that we aim to investigate in the future, are the levels of local versus systemic innate immune activation in response to treatment. CpG ODN trigger innate immune activation via TLR9, a pattern recognition receptor expressed by

plasmacytoid DC and B cells in humans and also found on myeloid immune cells in the mouse. Despite these differences between the human and the mouse immune system, the levels of systemic cytokine induction in the mouse induced by antibody-conjugated versus unconjugated CpG can provide evidence whether or not the adverse responses associated with systemic administration of TLR agonists can be curbed by the targeted approach. If systemic innate immune activation can be reduced while the local inflammatory microenvironment in the tumor tissue can be promoted to support the induction of cellular immunity and the functionality of immune effector function, the clinical application of targeted delivery of nucleic acid adjuvants would represent a further potent tool in the arsenal of tumor immunotherapeutic medicines.

## Supporting information

**S1 Fig. In vivo growth profiles of B14.3 HER2 tumor cell lines.** (A) Human HER2 expression of cultured B16 and B14.3 HER2 tumor cells was determined by flow cytometric analysis. (B) Lungs were harvested at day 15 post tumor inoculation. Upon fixation, the number of tumor nodules was counted, up to a maximum number of 250 nodules per mouse. (C) The expression of TLR3, TLR7 and TLR9 in B16 cells was analysed by PCR as describe previously by Edwards et al., 2003. TLR3, TLR7 and TLR9 encoding plasmids were used as control templates. Edwards AD, Diebold SS, Slack EM, Tomizawa H, Hemmi H, Kaisho T, et al. Toll-like receptor expression in murine DC subsets: lack of TLR7 expression by CD8 alpha+ DC correlates with unresponsiveness to imidazoquinolines. Eur J Immunol. 2003;33(4):827–33.
(TIF)

**S2 Fig. Pharmacokinetics of Trastuzumab-CpG conjugates in serum of treated mice. (A)** Schematic overview of the experiment. Mice bearing B14.3 HER2 tumors (n = 10) were treated 3 times with 40μg Trastuzumab with or without free CpG ODN (Tr +CpG or Tr, respectively), Trastuzumab-MCC-CpG (Tr-MCC-CpG), or TH-Trastuzumab-MCC-CpG (Tr(TH)-CpG). Serum samples were harvested at days 3 and 7 after the last round of treatment. **(B)** The serum concentration of human IgG1 was determined by sandwich ELISA using an antibody or respective conjugate standard curve. Data are shown as mean ± SD and are pooled from 3 independent experiments.
(TIF)

**S3 Fig. Gating strategy for flow cytometric analysis of splenic T cells.** (**A**) For analysis of different T cell populations, splenocytes from treated mice were harvested at day 18, stained LIVE/DEAD fixable dye and CD3-, CD4-, CD8-, CD44- and CD62L-specific antibodies and analyzed by flow cytometry. The gating strategy for identifying the percentage of CD44[+] CD62L[+] memory T cells for CD4[+] and CD8[+] T cell populations is shown. FMO controls were used for gating. **(B)** For antigen-specific proliferation assays, splenocytes were labelled with CFSE and cultured for 3 days with GM-CSF derived BMDC loaded with tumor specific proteins and peptides. Cells were harvested and stained for CD3, CD4, CD8, CD44 and analyzed by flow cytometry. FMO controls were used for gating. **(C)** The percentage of CFSE diluting cells was calculated and served as a marker of proliferation. Representative histograms of activated CD8[+] T cells from mice treated with TH-Trastuzumab (Tr(TH)) and TH-Trastuzumab-MCC-CpG conjugate (Tr(TH)-CpG) are shown as examples.
(TIF)

**S4 Fig. T cell numbers in spleens of mice upon repeated treatment with TH-Trastuzumab-MCC-CpG.** Mice were treated on days 4, 8 and 11 post tumor inoculation with 2mg/kg body weight of the indicated antibodies or antibody conjugates or with a dose of free CpG

equivalent to the CpG dose applied in conjugated form. Splenocytes were isolated on day 18. Splenocytes were stained and analyzed *ex vivo* by flow cytometry to determine the absolute counts of (A) CD4$^+$ versus CD8$^+$ CD3$^+$ T cells or (B) the percentage of CD62L$^+$ versus CD62L$^-$ CD44$^+$ cells within the CD4$^+$ and CD8$^+$ T cell populations. Each symbol represents a mouse treated with vehicle (n = 5), Iso-MCC-CpG, Tr-MCC-CpG, Tr(TH)-CpG or Tr (n = 8 for each), and Tr + CpG (n = 7). Means ± SD of samples from two independent experiments are depicted. Data were analyzed by Mann-Whitney two tailed test; $^*$p<0.05; $^{**}$p<0.01; ns— not significant.
(TIF)

**S1 Table. Overview over the different conditions used for generation of antibody-MCC-CpG conjugates.** Trastuzumab (Tr) and isotype control antibody (Iso) were conjugated to an excess of ODN and employing excess cross-linker SMCC at different ratios of CpG to antibody and SMCC to antibody as listed. The conjugation was performed in PBS or borate buffer as shown and the molar ratio of ODN to antibody was determined for each conjugate individually.
(DOCX)

**S1 Raw images.**
(PDF)

**S1 Dataset.**
(XLSX)

# Acknowledgments

We would like to thank the following staff members of the NIBSC Biological Service Division for their outstanding support and their excellent work in connection with this study: Shaun Baker, Rose Leahy, Alan Haynes, Luke Gurney, Lisa Johnson, Christine Zverev, Lauren Mackinnon and Arturo Fernandez. We would also like to thank Prof Sophia Karagiannis and Dr Sophie Papa at KCL for valuable support and advice provided to Diana Corogeanu during thesis progression meetings. In addition, we thank Dr Ram Abuknesha for his expert advice on the crosslinker cyanuric chloride.

# Author Contributions

**Conceptualization:** Andrew J. Beavil, James N. Arnold, Sandra S. Diebold.

**Data curation:** Diana Corogeanu, Kam Zaki.

**Formal analysis:** Diana Corogeanu.

**Investigation:** Diana Corogeanu, James N. Arnold.

**Methodology:** Diana Corogeanu, Kam Zaki, Andrew J. Beavil, James N. Arnold, Sandra S. Diebold.

**Resources:** Sandra S. Diebold.

**Supervision:** Andrew J. Beavil, James N. Arnold, Sandra S. Diebold.

**Writing – original draft:** Diana Corogeanu, Sandra S. Diebold.

**Writing – review & editing:** James N. Arnold, Sandra S. Diebold.

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
