## [Decision Letter · Decision Letter 0]

20 Jul 2022

PONE-D-22-14973Antibody conjugates for targeted delivery of Toll-like receptor 9 agonist to the tumor tissuePLOS ONE

Dear Dr. Diebold,

Thank you for submitting your manuscript to PLOS ONE. After careful consideration, we feel that it has merit but does not fully meet PLOS ONE’s publication criteria as it currently stands. Therefore, we invite you to submit a revised version of the manuscript that addresses the points raised during the review process.

We look forward to receiving your revised manuscript.

Kind regards,

Paulo Lee Ho, Ph.D.

Academic Editor

PLOS ONE

Journal Requirements:

2. To comply with PLOS ONE submissions requirements, in your Methods section, please provide additional information regarding the experiments involving animals and ensure you have included details on (1) methods of sacrifice, (2) methods of anesthesia and/or analgesia, (3) efforts to alleviate suffering and (4) tumor volume.

   "James N. Arnold is supported by grant from Cancer Research UK (DCRPGF\\100009) and is the recipient of a Cancer Research Institute / Wade F.B. Thompson CLIP grant (CRI3645)."

   "We would like to thank the following staff members of the NIBSC Biological Service Division for their outstanding support and their excellent work in connection with this study:Shaun Baker, Rose Leahy, Alan Haynes, Luke Gurney, Lisa Johnson, Christine Zverev,Lauren Mackinnon and Arturo Fernandez. We would also like to thank Prof Sophia Karagiannis and Dr Sophie Papa at KCL for valuable support and advice provided to Diana Corogeanu during thesis progression meetings. In addition, we thank Dr Ram Abuknesha for his expert advice on the crosslinker cyanuric chloride. J.N.A is supported by grant from Cancer Research UK (DCRPGF\\100009) and is the recipient of a Cancer Research Institute / Wade F.B. Thompson CLIP grant (CRI3645)"

 "James N. Arnold is supported by grant from Cancer Research UK (DCRPGF\\100009) and is the recipient of a Cancer Research Institute / Wade F.B. Thompson CLIP grant (CRI3645)."

7. We note that you have included the phrase “data not shown” in your manuscript. Unfortunately, this does not meet our data sharing requirements. PLOS does not permit references to inaccessible data. We require that authors provide all relevant data within the paper, Supporting Information files, or in an acceptable, public repository. Please add a citation to support this phrase or upload the data that corresponds with these findings to a stable repository (such as Figshare or Dryad) and provide and URLs, DOIs, or accession numbers that may be used to access these data. Or, if the data are not a core part of the research being presented in your study, we ask that you remove the phrase that refers to these data.

Reviewers' comments:

Reviewer's Responses to Questions

**Comments to the Author**

1. Is the manuscript technically sound, and do the data support the conclusions?

Reviewer #1: Yes

Reviewer #2: Yes

2. Has the statistical analysis been performed appropriately and rigorously? 

Reviewer #1: Yes

Reviewer #2: Yes

3. Have the authors made all data underlying the findings in their manuscript fully available?

Reviewer #1: Yes

Reviewer #2: Yes

4. Is the manuscript presented in an intelligible fashion and written in standard English?

Reviewer #1: Yes

Reviewer #2: Yes

5. Review Comments to the Author

Reviewer #1: This is a very comprehensive methods paper comparing heterobifunctional conjugation and site specific conjugation of TLR9 agonist CpG to Herceptin antibody. The methods and results are described clearly and show that both methods are effective in preserving the function and integrity of the immunoconjugate. This reviewer only has a few comments for the authors including:

1. While reading the manuscript, I got the impression that in vivo work would be included. However, after realizing it was not, I think it is better to consider changing the title to better reflect what is described by adding the following to the title:

"In Vitro Analysis of Antibody Conjugates for targeted delivery of Toll-like receptor...."

2. Although the investigators conclude that site directed conjugation is a preferred method over heterobifunctional conjugation procedures, both are actually good. In addition, they should point out that site directed conjugation requires sequencing the antibody and genetically engineering specific sites that do not interfere with antibody binding. Also, additional work will be needed to see the optimal number of sites that need to be changed to cysteines. With heterobifunctional conjugation, one can add 1,3, 6 etc CpG's and then demonstrate the best number needed for potency. Of course, not all molecules will have the same number of CpGs which is the strength of site directed methods. This needs to be pointed out.

3. The lack of in vivo data is disappointing especially since other prior investigators have generated such data (ref 42,43). Regarding this, a key point that was not stated is that CpG alone given intravenously or systemically is not effective due to its very short half life. Hence, in the past CpG was only given intra-tumorally until it was eventually conjugated to tumor targeting antibodies. No question that targeted CpG is superior but it is still not clear what cells the CpG should be targeted to. It could be intra-tumor, intravascular, or targeted to antigen presenting cells in lymph nodes or resident in tumors. This point also needs to be addressed.

4. For in vivo studies, it is best to use syngeneic tumor models and antibodies that target tumors of these models. As described by the investigators, herceptin and other human specific targets make the tumor more immunogenic in animal models to mask the immune effects of the immunoconjugate alone. This could be a point for the discussion.

5.Curiously, there is little discussion of how CpG or Toll-like agonists arm the immune system. This would make the paper much more interesting in the Introduction. In addition, it is not obvious to most researchers that TLR9 receptors are not on the surface of antigen presenting cells but are located in the endosomes in the cytoplasms. I believe because of this, most investigators did not try these agonists for cancer immunotherapy. However, in ref 43, co-locatlization studies clearly show that the CpG is delivered internally to the TLR9 receptor in the endosomes, while the antibody remains on the cell surface of APCs.

Reviewer #2: The manuscript by Corogeanu et al reports on the generation of specific Trastuzumab antibody CpG conjugates for cancer immunotherapy. Overall the experiments are well-designed and performed. Novelty of the studies is limited, however, and the manuscript is quite lengthy and could be written in a more concise manner.

Questions/remarks

- Fig 2. The amount of 0.25 ug of CpG appears a stronger activator than double the amount of CpG (0.5 ug). This is unexpected and probably not correct, please explain.

- Fig 3 ABC show somewhat less inhibition of proliferation by the conjugates relative to Tratuzumab, but the differences are minor especially with the scale starting at 50 %. The difference with TH-conjugate in fig 5. are also relatively small. This appears quite different from the binding to immobilized Her2 protein in Fig 4. Has FACS staining been done with the different mAbs to detect actual binding to cells in stead of her2 protein?

Fig 4. Use differences in binding to immobilized Her2 protein are observed. This is not in line with the proliferation data. Can the authors exclude that binding of the detection mAb used is affected by conjugation rather then the binding to Her2 protein?

- The data are quite specific for Trastuzumab of which the AA sequence is known. The manuscript could be improved by for example adding info on the location and number of the targeted AA in the conjugation process. Something special?

-6D, why are the data for the regular Trastuzumab-CpG conjugates not shown in this figure? Please add.

- Overall the manuscript is quite lengthy, please shorten. Certain partners of the reults section belong more to the M&M.

- The murine models are quite artificial both from tumor (no direct effect on B16) as the immune component and results appear quite variable within groups Could the authors comment on the variation in the groups? In my opinion the addition of all the immunomonitoring in this model is a bit difficult to support as the model is so artificial. Maybe better to shorten and focus on the conjugate characterization parts.

6. PLOS authors have the option to publish the peer review history of their article (what does this mean?). If published, this will include your full peer review and any attached files.

Reviewer #1: **Yes: **Alan Epstein MD, PhD

Reviewer #2: No

---

## [Author Response · Author response to Decision Letter 0]

9 Oct 2022

We would like to thank you for giving us the opportunity to submit a revised version of our manuscript entitled “Antibody conjugates for targeted delivery of Toll-like receptor 9 agonist to the tumor tissue”.

We very much appreciate the positive and helpful comments both reviewers made about the manuscript. We have revised the manuscript according to their suggestions where possible as outlined below in our point-by-point response. We also have addressed the additional journal requirements as requested.

We hope that the revised version of our manuscript is acceptable for publication in PLOS ONE and are looking forward to hearing from you.

Point-by-point response to the reviewer’s comments:

(1) Addressing the Journal Requirements:

1. PLOS ONE’s style requirement

We have revised the file names according to the PLOS ONE style requirements.

2. Revision of Method Section

The revised method section describing the tumor model now includes additional information on (1) methods of sacrifice, (2) methods of anesthesia and/or analgesia, (3) efforts to alleviate suffering and (4) tumor volume as requested (page 12 line 331-338). The tumor volume could not be evaluated in this pseudo-metastasis model which generates internal tumor nodules in the lung.

3. Revision of Financial Disclosure

The following financial disclosure regarding the funding is now included in the revised cover letter: James Arnold is supported by a grant from Cancer Research UK (DCRPGF\\100009) and is the recipient of a Cancer Research Institute / Wade F.B. Thompson CLIP grant (CRI3645). A sentence to clarify the role the funder took in the study is also included in the cover letter as requested.

4. Revision of Acknowledgement Section

The information on funding was removed from the Acknowledgement section. The funding statement is now included in the revised cover letter (see point 3. above) and, also, includes clarification regarding the role the funders took in the study. Please amend our online funding information to include the role of the funders accordingly.

5. Revision of Data Availability

We have revised the online data availability statement as requested and will provide all data without restrictions. 

6. Supporting Information for Gel Results

We have assembled the original complete gel photographs in a supplementary file as requested.

7. Revision Regarding “data not shown”

The text was revised accordingly and we removed the references to data not shown.

(2) Addressing the Comments from Reviewer 1:

2.1. In Vivo Work

Reviewer 1 wrote: While reading the manuscript, I got the impression that in vivo work would be included. However, after realizing it was not, I think it is better to consider changing the title to better reflect what is described by adding the following to the title:

"In Vitro Analysis of Antibody Conjugates for targeted delivery of Toll-like receptor...."

Figures 6 and 7 describe the results from the mouse tumor model. Since in vivo work is included in the manuscript we assume that it is acceptable to leave the title of the manuscript unchanged. 

2.2. Stochastic Versus Site-Specific Conjugation

Although the investigators conclude that site directed conjugation is a preferred method over heterobifunctional conjugation procedures, both are actually good. In addition, they should point out that site directed conjugation requires sequencing the antibody and genetically engineering specific sites that do not interfere with antibody binding. Also, additional work will be needed to see the optimal number of sites that need to be changed to cysteines. With heterobifunctional conjugation, one can add 1,3, 6 etc CpG's and then demonstrate the best number needed for potency. Of course, not all molecules will have the same number of CpGs which is the strength of site directed methods. This needs to be pointed out.

We have addressed the reviewer’s comment on stochastic versus site-specific conjugation by revising the manuscript as requested. In the starting paragraph on site-specific conjugate generation we have revised the manuscript to point out that site-specific ThioMab technology “involves genetically engineering cysteines at specific sites in the structure of Trastuzumab that do not interfere with antigen binding“ (page 22, lines 562-563).

We also introduced a paragraph in the discussion to point out that future work is required to identify the optimal number of engineered cysteines for site-specific conjugation (page 32, lines 809-815): “While we generated stochastic conjugates with various CpG to antibody ratios from 0.4:1 to 9.16:1, the ratio of CpG to antibody for site-specific conjugates was limited by the introduction of only one engineered cysteine site at position 205 in the antibody light chain resulting in conjugates with 1.5 to 1.9 CpG molecules per antibody molecule. Future studies will have to clarify whether the introduction of additional engineered cysteine moieties and an increase in the payload of site-specific conjugates is of therapeutic benefit.”

The strength of the site-specific conjugates is addressed in the starting paragraph on site-specific conjugate generation (page 21, line 559-561): “In addition to unaltered antigen binding properties, site-specific conjugates promise higher homogeneity and in vivo stability, influencing both safety and therapeutic efficacy of conjugates.”

2.3. Targeted delivery of CpG 

Regarding this, a key point that was not stated is that CpG alone given intravenously or systemically is not effective due to its very short half life. Hence, in the past CpG was only given intra-tumorally until it was eventually conjugated to tumor targeting antibodies. No question that targeted CpG is superior but it is still not clear what cells the CpG should be targeted to. It could be intra-tumor, intravascular, or targeted to antigen presenting cells in lymph nodes or resident in tumors. This point also needs to be addressed.

In response to this comment from reviewer 1, we have added a paragraph in the discussion (pages 31-32, lines 819-826): “Targeting TLR9 agonist to the tumor tissue is thought to enable the priming of antigen-specific T cells in the draining lymph nodes, to promote the tumor infiltration with effector cells and to disable immune checkpoints in the tumor tissue leading to efficient anti-tumor responses. While alternative approaches such as targeting antigen-presenting cells may be more effective in promoting T cell priming, the use of a therapeutic tumor-targeting antibody such as Trastuzumab has the added potential benefit to boost synergistic effects between the TLR agonist and the therapeutic antibody.”

2.4. Human Specific Targets in the Mouse Model

For in vivo studies, it is best to use syngeneic tumor models and antibodies that target tumors of these models. As described by the investigators, herceptin and other human specific targets make the tumor more immunogenic in animal models to mask the immune effects of the immunoconjugate alone. This could be a point for the discussion.

The advantages and disadvantages of the used syngeneic C57BL/6 mouse model with transgenic tumor cells are discussed extensively in the discussion of the manuscript (page 32, lines 828-867). In lines 837-839 we address the fact that human HER2 is a de novo antigen in this mouse model: “The most prominent of these caveats is the fact that human HER2 represents a foreign antigen in mice. The expression of a foreign antigen is the likely reason for a noticeable reduction in tumor burden for B14.3 HER2 cells when compared to the parental HER2-negative cells.” 

In the second paragraph on this subject, we also discuss that humanized mouse model with a human tumor xenograft may be better suited for investigation of certain aspects of the conjugates. However, like all tumor models, even the humanized mouse xenograft tumor model is far from perfect and has caveats which will affect the conclusions one can draw from this model.

2.5. Recognition of TLR9 agonists in the Endosomal Compartment

Curiously, there is little discussion of how CpG or Toll-like agonists arm the immune system. This would make the paper much more interesting in the Introduction. In addition, it is not obvious to most researchers that TLR9 receptors are not on the surface of antigen presenting cells but are located in the endosomes in the cytoplasms. I believe because of this, most investigators did not try these agonists for cancer immunotherapy. However, in ref 43, co-locatlization studies clearly show that the CpG is delivered internally to the TLR9 receptor in the endosomes, while the antibody remains on the cell surface of APCs.

In response to this comment from reviewer 1, we have revised the first paragraph of the introduction (page 3, lines 50-59). The text now reads as follows: “TLR3, TLR7/8 and TLR9 are pattern recognition receptors that reside intracellularly, in the endosome of innate immune cells including antigen-presenting cells such as dendritic cells (DC) and macrophages. These endosomal TLR detect nucleic acids from viruses and other intracellular pathogens and mediate innate immune activation. Upon activation of DC by these endosomal TLR, the cells undergo maturation, migrate to the draining lymph nodes and present exogenous antigens to CD4 and CD8 T cells. Importantly, the nature of the initial stimulus shapes the phenotype of the induced T cell response. DC activated by the detection of nucleic acid agonists via the endosomal TLR instruct adaptive immune responses towards a T helper 1 (Th1) phenotype with strong cytotoxic T cell (CTL) effector functions for elimination of infected cells.”

Please note: TLR9 agonists have been tested extensively in the context of tumor therapy in the clinic and numerous reviews have been written about the subject. There are numerous reasons why TLR9 agonist monotherapy does not achieve the desired efficacy in the clinical setting. It goes beyond the capacity of our manuscript to list and discuss all the evidence. However, from past trials it was widely concluded that TLR9 agonists are suitable adjuvants for induction of anti-tumor CTL responses, but that their clinical application needs refinement and potentially combination with other therapeutic approaches. Our study represents such a refined combinatorial approach.

(3) Addressing the Comments from Reviewer 2:

3.1. Dose Response for TLR9 Agonist CpG

Fig 2. The amount of 0.25 ug of CpG appears a stronger activator than double the amount of CpG (0.5 ug). This is unexpected and probably not correct, please explain.

We thank reviewer 2 for bringing up this point. At closer inspection when compiling the raw data sets for submission, we realized that the two concentrations on the figure were accidentally swapped. We have now rectified this in the revised Figure 2. Unfortunately, we did not spot this, since many of the TLR agonists including CpG show a bell-shaped dose response curve for inflammatory cytokines such as IL-6 and IL-12. At higher levels of the TLR agonist increasing levels of IL-10 are induced. The increased levels of this anti-inflammatory cytokine lead to reduced production of pro-inflammatory cytokines in in vitro assay because of a negative regulatory feedback loop.

3.2. HER2 Binding Assay

Fig 3 ABC show somewhat less inhibition of proliferation by the conjugates relative to Tratuzumab, but the differences are minor especially with the scale starting at 50 %. The difference with TH-conjugate in fig 5. are also relatively small. This appears quite different from the binding to immobilized Her2 protein in Fig 4. Has FACS staining been done with the different mAbs to detect actual binding to cells instead of her2 protein?

The growth inhibiting activity of Trastuzumab does correlate with its HER2 binding activity, but the two assays are not expected to “behave” exactly in the same way. Inhibition of proliferation of tumour cells in vitro requires high concentration of Trastuzumab and this effect saturates at a level of tumor growth inhibition of around 60% at around 100nM of antibody for the two Trastuzumab-sensitive tumor cell lines we have tested (Fig. 3). Higher levels of tumor growth inhibition cannot be achieved in this in vitro assay. 

The HER2 binding assay is more sensitive. The binding of Trastuzumab is saturated between 1-3nM of the antibody (Fig.4). The HER2 binding assay was performed with different detection antibodies as stated in the material & methods section. The two detection antibodies gave comparable results. 

We have undertaken flow cytometric analysis of mAb binding to cells, but have not included this in the manuscript, since it did not provide more insight into HER2 binding. The HER2 binding assay provided better quantitative data allowing us to better distinguish between conjugates with different PARs and enabling us to show clear dose responses.

3.3. Binding of Detection Antibody by Conjugates

Fig 4. Use differences in binding to immobilized Her2 protein are observed. This is not in line with the proliferation data. Can the authors exclude that binding of the detection mAb used is affected by conjugation rather than the binding to Her2 protein?

Please see answer to point 3.2 above. Detection antibodies against the Fc region of human IgG1 and against the human kappa light chain were used and gave comparable results. This reassured us that the conjugation did not negatively affect the detection of the conjugates and that the assay results correctly reflect the level of HER2 binding by the conjugates.

3.4. Antibody AA Sequence

The data are quite specific for Trastuzumab of which the AA sequence is known. The manuscript could be improved by for example adding info on the location and number of the targeted AA in the conjugation process. Something special?

In response to this comment, we have revised the manuscript and added a sentence supported by two references in the discussion on page 30, lines 788-790: “There are 88 lysine groups in Trastuzumab which represent available solvent NH2 groups amenable to SMCC-mediated conjugation which give rise to highly heterogenous populations of conjugates.” 

3.5. Stochastic Trastuzumab-CpG Conjugates

6D, why are the data for the regular Trastuzumab-CpG conjugates not shown in this figure? Please add.

Unfortunately, we took only photographs of the groups shown in Figure 6D. We did not take photographs of the lungs from the control groups (vehicle, Iso-MCC-CpG) or the Tr-MCC-CpG group. Our rational at the time was that the level in tumor burden in these groups as shown in Figure 6C was not different from the Tr and the CpG control groups hence the lungs looked the same. In figure 6D, we aimed to specifically show the reduced tumor burden in the Tr(TH)-CpG and Tr +CpG group.

Since we did not take photographs of the lungs from Tr-MCC-CpG treated mice, we are unable to add them to figure 6D.

3.6. Length of Manuscript

Overall, the manuscript is quite lengthy, please shorten. Certain partners of the reults section belong more to the M&M.

We agree with reviewer 2 that the manuscript is quite lengthy and very technical. However, since the aim of the study was to compare various conjugation strategies, these technical aspects constitute results rather than material and methods. 

In order to shorten the manuscript, we removed a sentence on page 21 which we felt did not add to a better understanding of the study. However, we felt unable to identify further superfluous text passages. Moreover, both reviewers requested further clarifications and discussion points to be included in the manuscript.

---

## [Decision Letter · Decision Letter 1]

11 Jan 2023

PONE-D-22-14973R1Antibody conjugates for targeted delivery of Toll-like receptor 9 agonist to the tumor tissuePLOS ONE

Dear Dr. Diebold,

Thank you for submitting your manuscript to PLOS ONE. After careful consideration, we feel that it has merit but does not fully meet PLOS ONE’s publication criteria as it currently stands. Therefore, we invite you to submit a revised version of the manuscript that addresses the points raised during the review process.

We look forward to receiving your revised manuscript.

Kind regards,

Paulo Lee Ho, Ph.D.

Academic Editor

PLOS ONE

Journal Requirements:

Reviewers' comments:

Reviewer's Responses to Questions

**Comments to the Author**

1. If the authors have adequately addressed your comments raised in a previous round of review and you feel that this manuscript is now acceptable for publication, you may indicate that here to bypass the “Comments to the Author” section, enter your conflict of interest statement in the “Confidential to Editor” section, and submit your "Accept" recommendation.

Reviewer #1: (No Response)

Reviewer #2: (No Response)

2. Is the manuscript technically sound, and do the data support the conclusions?

Reviewer #1: (No Response)

Reviewer #2: Partly

3. Has the statistical analysis been performed appropriately and rigorously? 

Reviewer #1: (No Response)

Reviewer #2: Yes

4. Have the authors made all data underlying the findings in their manuscript fully available?

Reviewer #1: (No Response)

Reviewer #2: Yes

5. Is the manuscript presented in an intelligible fashion and written in standard English?

Reviewer #1: (No Response)

Reviewer #2: Yes

6. Review Comments to the Author

Reviewer #1: (No Response)

Reviewer #2: The authors have made some effort to improve and shorten the manuscript.

Remaining issues:

3.2. HER2 Binding Assay

“We have undertaken flow cytometric analysis of mAb binding to cells, but have not included this in the manuscript, since it did not provide more insight into HER2 binding”.

Response: The authors do not have to show the data, but they should mention the outcome of the experiment at least in the rebuttal. Were the FACS stainings identical?

3.6. Length of Manuscript

In order to shorten the manuscript, we removed a sentence on page 21 which we felt did not add to a better understanding of the study. However, we felt unable to identify further superfluous text passages. Moreover, both reviewers requested further clarifications and discussion points to be included in the manuscript.

Response:This is really a minor shortening. The last reviewer comment concerning the murine models and immunomonitoring (see below) was not answered by the authors and included some suggestions to shorten.

The murine models are quite artificial both from tumor (no direct effect on B16) as the immune component and results appear quite variable within groups Could the authors comment on the variation in the groups? In my opinion the addition of all the immunomonitoring in this model is a bit difficult to support as the model is so artificial. Maybe better to shorten and focus on the conjugate characterization parts.

7. PLOS authors have the option to publish the peer review history of their article (what does this mean?). If published, this will include your full peer review and any attached files.

Reviewer #1: No

Reviewer #2: No

---

## [Author Response · Author response to Decision Letter 1]

22 Feb 2023

6. Review Comments to the Author

Reviewer #1: (No Response)

Reviewer #2: The authors have made some effort to improve and shorten the manuscript.

Remaining issues:

3.2. HER2 Binding Assay

“We have undertaken flow cytometric analysis of mAb binding to cells, but have not included this in the manuscript, since it did not provide more insight into HER2 binding”.

Response: The authors do not have to show the data, but they should mention the outcome of the experiment at least in the rebuttal. Were the FACS stainings identical?

Authors’ Response: Please see attachment for a response to this reviewer comment. We include a representative figure of the FACS staining, and also a brief explanation of the results. The flow cytometry assay supports the conclusions drawn from the HER2 binding assay, however, we had concerns that minor differences in binding of different stochastic conjugates that we observed in the HER2 binding assay were not reliably quantifiable in the flow cytometry assay.

Please Note: The attachment is included in the submission for review by the reviewers only and is not meant to be included as supplementary information as part of the manuscript.

3.6. Length of Manuscript

In order to shorten the manuscript, we removed a sentence on page 21 which we felt did not add to a better understanding of the study. However, we felt unable to identify further superfluous text passages. Moreover, both reviewers requested further clarifications and discussion points to be included in the manuscript.

Response: This is really a minor shortening. The last reviewer comment concerning the murine models and immunomonitoring (see below) was not answered by the authors and included some suggestions to shorten.

The murine models are quite artificial both from tumor (no direct effect on B16) as the immune component and results appear quite variable within groups Could the authors comment on the variation in the groups? In my opinion the addition of all the immunomonitoring in this model is a bit difficult to support as the model is so artificial. Maybe better to shorten and focus on the conjugate characterization parts.

Authors’ Response: Yes, reviewer 1 was correct in pointing out that the groups are very variable. The high variability was seen in all experiments and in all treatment groups and the reason for this high variability is currently unknown. Injections were performed by highly trained and experienced animal technicians, and we do not suspect poor injection technique to play a role in the high variability observed. One possible explanation is that the human HER2-expressing modified melanoma cells are inherently more immunogenic than the parental cell line and that in a fraction of mice immune responses directed against the human neoantigen develop spontaneously. We now mention this in the discussion of the revised version of the manuscript on page 32.

While the model is artificial, the results are significant despite the high variability within groups. In our discussion, we have been open about the pros and cons of this model and, also discuss the pros and cons of alternative models such as the humanised model. To our knowledge there is no ideal animal model to study the function of a therapeutic antibody such as Trastuzumab targeting a human tumour marker. Humanised mouse models are also highly artificial and have their own caveats. 

In order to comply with the reviewers’ request to shorten the manuscript, we have revised the manuscript further by simplifying the introduction of the tumour model on page 24 and by editing the presentation of the results on pages 25-28. We also revised the discussion of the tumour model on pages 32-33 in order to shorten the manuscript further. Furthermore, we have made deletions in other parts of the text and have moved a paragraph from the result section to the materials & method section. We were anxious to not delete any relevant observations and discussion points regarding the tumour model from the manuscript. We hope that this level of shortening of the manuscript is acceptable and we would like to point out that PLOS ONE does not have restrictions on the length of articles.

---

## [Editor Report · Decision Letter 2]

24 Feb 2023

Antibody conjugates for targeted delivery of Toll-like receptor 9 agonist to the tumor tissue

PONE-D-22-14973R2

Dear Dr. Diebold,

We’re pleased to inform you that your manuscript has been judged scientifically suitable for publication and will be formally accepted for publication once it meets all outstanding technical requirements.

Kind regards,

Paulo Lee Ho, Ph.D.

Academic Editor

PLOS ONE
---

## [Editor Report · Acceptance letter]

3 Mar 2023

PONE-D-22-14973R2 

Antibody conjugates for targeted delivery of Toll-like receptor 9 agonist to the tumor tissue 

Dear Dr. Diebold:

I'm pleased to inform you that your manuscript has been deemed suitable for publication in PLOS ONE. Congratulations! Your manuscript is now with our production department. 

Kind regards, 

on behalf of

Dr. Paulo Lee Ho 

Academic Editor

PLOS ONE